# Inequality in infrastructure access and its association with health disparities

Ying Tu [1,2,3,4], Bin Chen [2,5] ✉, Chuan Liao [3] ✉, Shengbiao Wu [2,5], Jiafu An [5,6], Chen Lin [5,7], Peng Gong [5,8], Bin Chen [9], Hong Wei [1] & Bing Xu [1,4] ✉

Economic, social and environmental infrastructure forms a fundamental pillar of societal development. Ensuring equitable access to infrastructure for all residents is crucial for achieving the Sustainable Development Goals, yet knowledge gaps remain in infrastructure accessibility and inequality and their associations with human health. Here we generate gridded maps of economic, social and environmental infrastructure distribution and apply population-weighted exposure models and mixed-effects regressions to investigate differences in population access to infrastructure and their health implications across 166 countries. The results reveal contrasting inequalities in infrastructure access across regions and infrastructure types. Global South countries experience only 50–80% of the infrastructure access of Global North countries, whereas their associated inequality levels are 9–44% higher. Both infrastructure access and inequality are linked to health outcomes, with this relationship being especially pronounced in economic infrastructure. These findings underscore the necessity of informed decision-making to rectify infrastructure disparities for promoting human well-being.

Infrastructure systems—comprising various components such as transportation networks, energy supply systems, water and sanitation facilities—provide essential services that underpin socio-economic functioning and human well-being[1,2]. Understanding the distribution of and access to infrastructure, and the inequality therein, is of great importance to facilitate sustainable development and public health[3–7]. The United Nations has highlighted the need for "building resilient infrastructure, promoting sustainable industrialization, and fostering innovation" in its 9th Sustainable Development Goal (SDG 9)[8]. Notably, 72% of the 169 SDG targets have a direct association with infrastructure[4], and various forms of infrastructure can influence human health[9]. This close nexus between infrastructure, sustainability and health

underscores the need for research on how disparities in infrastructure access correlate with health outcomes among populations, which will inform targeted strategies for policymakers to improve infrastructure equity, optimize resource distribution, and promote sustainable development and human well-being.

Despite its importance, considerable knowledge gaps persist in the global landscape of infrastructure inequality and its health implications. One key challenge lies in the accurate and dynamic quantification of the amount, coverage and efficacy of infrastructure. Existing assessments of infrastructure investments and supplies primarily depend on socio-economic indicators or tangible assets, such as capital flows and stocks[10,11], urban built-up areas[12,13], and road networks[14]. However,

[1]Department of Earth System Science, Ministry of Education Ecological Field Station for East Asian Migratory Birds, Institute for Global Change Studies, Tsinghua University, Beijing, China. [2]Future Urbanity & Sustainable Environment (FUSE) Lab, Division of Landscape Architecture, Faculty of Architecture, The University of Hong Kong, Hong Kong SAR, China. [3]Department of Global Development, Cornell University, Ithaca, NY, USA. [4]International Research Center of Big Data for Sustainable Development Goals, Beijing, China. [5]Institute for Climate and Carbon Neutrality, Urban Systems Institute, The University of Hong Kong, Hong Kong SAR, China. [6]Department of Real Estate and Construction, Faculty of Architecture, The University of Hong Kong, Hong Kong SAR, China. [7]Faculty of Business and Economics, The University of Hong Kong, Hong Kong SAR, China. [8]Department of Geography and Department of Earth Sciences, The University of Hong Kong, Hong Kong SAR, China. [9]School of Environment, Beijing Normal University, Beijing, China. ✉e-mail: binley.chen@hku.hk; cl824@cornell.edu; bingxu@tsinghua.edu.cn

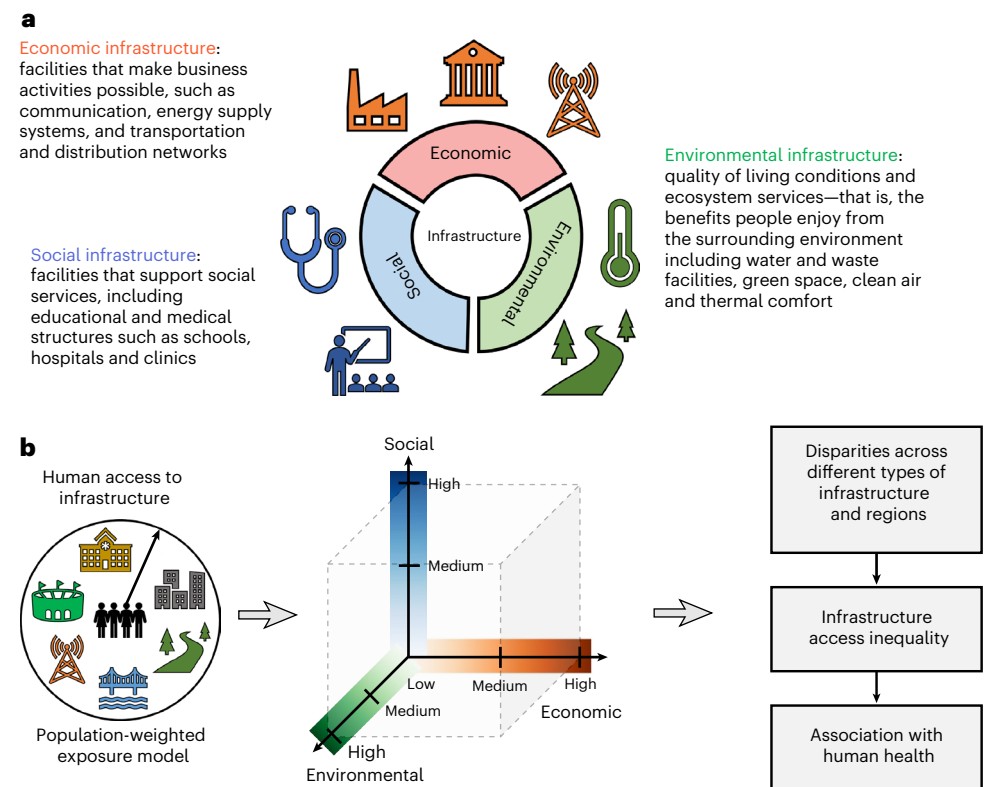

**Fig. 1 | Conceptual framework and research design. a**, Definition and classification of economic, social and environmental infrastructure in this study. **b**, Overview of the research design.

these methods, which assume uniform exposure across populations over time and space, tend to disregard the differences in individuals' access to nearby infrastructure. Consequently, cumulative unit measures of overall infrastructure supply or per capita supply can lead to biased knowledge about individual access when based on aggregated data[15,16]. Additionally, current evaluation methods largely focus on a single infrastructure system (for example, coastal built assets) and are often conducted at the regional scale[17–20], leading to diverging results on infrastructure access. Such approaches hinder a holistic understanding of infrastructure performance and prevent the identification of targeted decision-making and interventions to address infrastructure deficiencies. For reliable cross-regional comparisons, it is crucial to assess infrastructure accessibility across different types of infrastructure and at multiple scales, which requires using harmonized data sources derived from global-level mapping endeavours and assessment methods.

In recent years, concerns have grown regarding the adverse effects of infrastructure development, including ecological complications[21], heightened inequalities[22] and disruptions to sociocultural norms[23]. For example, large-scale infrastructure projects, such as road construction, can lead to substantial forest loss[24]. In urban areas, uneven spatial distribution of infrastructure and access to amenities can exacerbate inequalities among various social groups[25]. A number of recent studies conducted in India[26,27], Indonesia[28,29], South Africa[30,31] and elsewhere[32–35] have demonstrated disparities in infrastructure distribution and accessibility—such as electricity and telecommunications—across cities and communities, posing potential threats to urban sustainability and public health[3,6,36,37]. However, previous studies of infrastructure inequality have predominantly focused on evaluating a single dimension (for example, inequality in water infrastructure), the geographic extent of selected cities or the measurement scale of infrastructure-related accessibility indices[12,26–35], leaving large uncertainties in the spatially

explicit assessment of unequal access to different types of infrastructure on the global scale, particularly the differences between countries in the Global North and Global South.

Infrastructure systems can affect human health and well-being through multiple pathways[38]. For instance, energy supply outages after extreme events are associated with increased mortality risk[39,40]. Transportation infrastructure also affects health, with some impacts being beneficial (for example, providing physical connectivity) and others detrimental (for example, traffic noise and injury risks)[41]. Social infrastructure, such as health-care facilities, is positively linked to public health[42,43], and environmental infrastructure such as green spaces is widely seen as beneficial for physical and mental health[44,45]. Understanding how disparities in infrastructure access translate into health inequality is crucial, as infrastructure inequities often reflect and reinforce broader health disparities[36,46,47]. Recent research has demonstrated that inequities in access to health care and water infrastructure can lead to increased health burdens[48,49]. Nonetheless, most studies examining the relationship between infrastructure access and human health focus on single infrastructure types, be they economic, social or environmental, while research on health disparities across multiple infrastructure types often remains confined to regional scales[46,50].

To bridge these knowledge gaps, this paper aims to provide a comprehensive, multi-dimensional analysis of global inequality in infrastructure access and the associated disparity in health outcomes. Given the broad interpretation of infrastructure (see discussions on its definition and classification in Supplementary Information section 1), here we define 'infrastructure' as the physical systems and facilities essential for the functioning of the economy and society, 'infrastructure access' as individuals' capacity to reach and use nearby infrastructure, and 'infrastructure access inequality' as geographical disparities in the availability of these systems and facilities. We categorize infrastructure into three types based on its primary function:

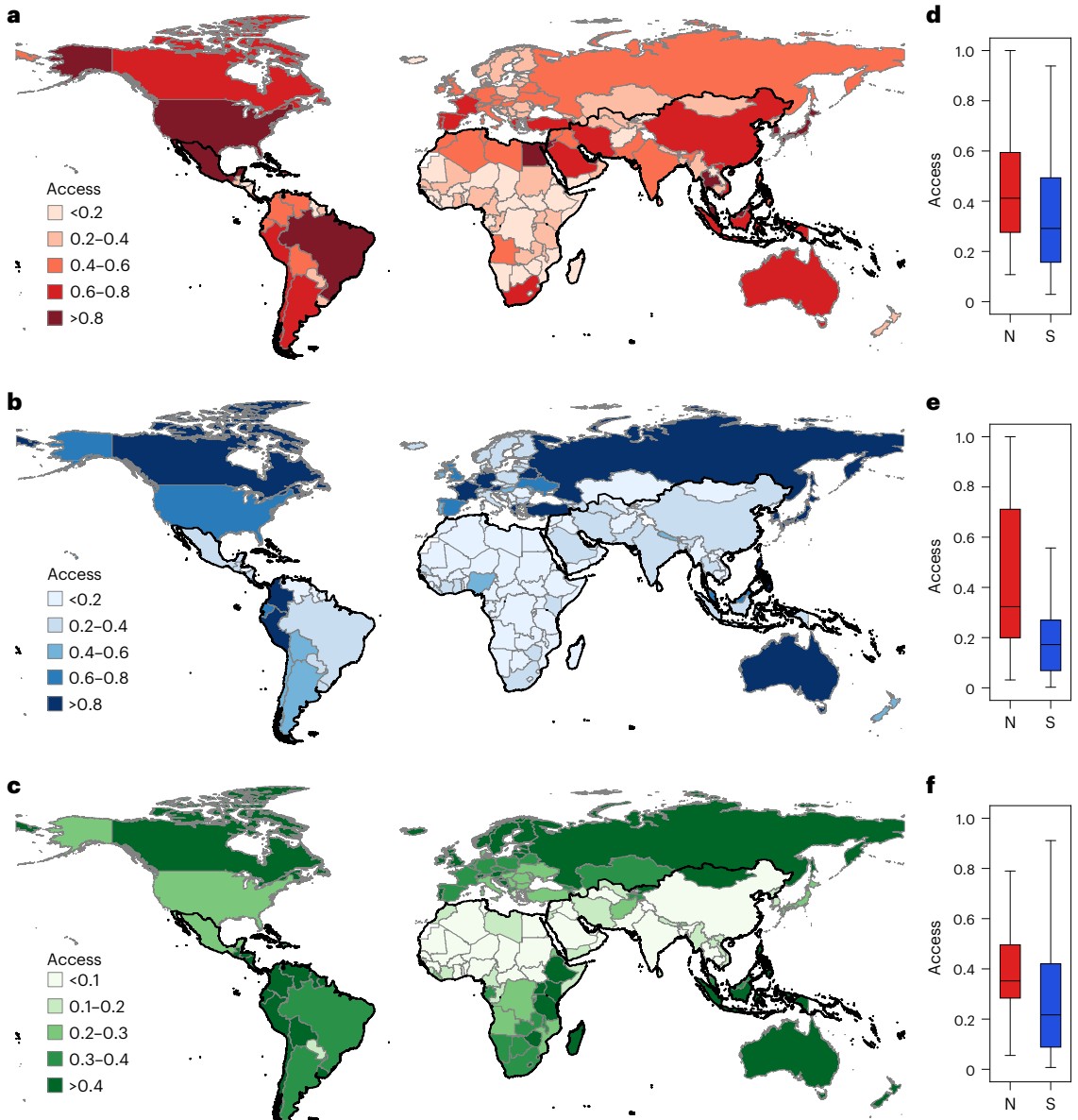

**Fig. 2 | Country-level distributions of human access to infrastructure.**
**a**–**c**, Global maps showing economic (**a**), social (**b**) and environmental (**c**) infrastructure access. Black boundaries delineate countries in the Global South. **d**–**f**, Box plots illustrating economic (**d**), social (**e**) and environmental (**f**) infrastructure access in countries of the Global North (N) and Global South (S). The box plots display the distribution of the data, with the median (50th percentile) at the centre, the interquartile range (25th to 75th percentiles) as the box, and whiskers extending to the maximum and minimum values within

1.5 times the interquartile range. The sample sizes were 54 countries for the Global North and 112 countries for the Global South. No replicates were carried out, and no adjustments for multiple comparisons were applied. Statistical significance was determined using a two-sided $t$-test for comparisons between the Global North and Global South (economic: $t_{164} = 2.14$; $P = 0.034$; Cohen's $d = 0.36$; 95% CI, (0.01, 0.17); social: $t_{164} = 5.43$; $P < 0.001$; Cohen's $d = 0.84$; 95% CI, (0.13, 0.32); environmental: $t_{164} = 2.99$; $P = 0.003$; Cohen's $d = 0.50$; 95% CI, (0.05, 0.22)).

economic, social and environmental (Fig. 1a). Economic infrastructure encompasses facilities that support business activities, such as telecommunication, energy supply systems, and transport and distribution networks. Social infrastructure comprises facilities providing social services, such as schools and hospitals. Environmental infrastructure, linked to living conditions and ecosystem services, includes resources such as water and waste facilities, green spaces, clean air, and thermal comfort. This multi-dimensional conceptualization of infrastructure leads to an enhanced understanding of the diverse ways in which individuals meet their needs and communities manage public goods across regions.

On the basis of this framework, we generated global maps of economic, social and environmental infrastructure for 2020 at 0.1° × 0.1°

spatial resolution by integrating multi-source geospatial datasets. We then employed a population-weighted exposure model to quantify human access to infrastructure and evaluated the associated inequality levels. Lastly, we examined the relationships between infrastructure access, infrastructure access inequality and human health outcomes. A detailed flow chart of the research design is presented in Supplementary Fig. 1. Specifically, we address three research questions: (1) What are the differences in infrastructure access across different types (economic, social and environmental) and scales (country and county levels)? (2) How do inequalities in infrastructure access vary across infrastructure types and regions? (3) What is the relationship between infrastructure access, associated inequality and human health outcomes?

**Table 1 | Statistics of country-level human access to economic, social and environmental infrastructure and infrastructure inequalities (measured by the Gini coefficient) across regions**

| Region | Count | Infrastructure access | | | Infrastructure access inequality (Gini) | | |
|---|---|---|---|---|---|---|---|
| | | Economic | Social | Environmental | Economic | Social | Environmental |
| Global North | 54 | 0.45±0.23 | 0.44±0.32 | 0.43±0.26 | 0.47±0.11 | 0.78±0.15 | 0.27±0.11 |
| Global South | 112 | 0.36±0.26 | 0.22±0.21 | 0.30±0.27 | 0.58±0.11 | 0.85±0.10 | 0.39±0.13 |
| Europe | 40 | 0.38±0.15 | 0.39±0.29 | 0.47±0.25 | 0.46±0.09 | 0.81±0.13 | 0.25±0.09 |
| Asia | 46 | 0.53±0.30 | 0.30±0.26 | 0.21±0.21 | 0.54±0.13 | 0.81±0.16 | 0.43±0.14 |
| North America | 16 | 0.41±0.27 | 0.35±0.29 | 0.44±0.23 | 0.54±0.10 | 0.80±0.11 | 0.38±0.10 |
| South America | 12 | 0.48±0.24 | 0.44±0.33 | 0.56±0.23 | 0.59±0.05 | 0.82±0.06 | 0.41±0.10 |
| Oceania | 4 | 0.32±0.31 | 0.34±0.38 | 0.95±0.11 | 0.49±0.07 | 0.73±0.08 | 0.27±0.10 |
| Africa | 48 | 0.24±0.18 | 0.13±0.11 | 0.24±0.22 | 0.61±0.12 | 0.87±0.09 | 0.34±0.12 |
| Global | 166 | 0.39±0.25 | 0.29±0.27 | 0.35±0.27 | 0.55±0.12 | 0.83±0.12 | 0.35±0.14 |
| One-way ANOVA across infrastructure types | | $F_{2,163}$=6.08; $P$=0.002; Cohen's $f$=0.87; 95% CI, (5.71, 6.46) | | | $F_{2,163}$=581.82; $P$<0.001; Cohen's $f$=1.00; 95% CI, (578.16, 585.49) | | |
| One-way ANOVA across regions | | $F_{5,160}$=7.89; $P$<0.001; Cohen's $f$=0.78; 95% CI, (7.08, 8.71) | $F_{5,160}$=6.40; $P$<0.001; Cohen's $f$=0.75; 95% CI, (5.66, 7.13) | $F_{5,160}$=16.11; $P$<0.001; Cohen's $f$=0.87; 95% CI, (14.95, 17.27) | $F_{5,160}$=8.53; $P$<0.001; Cohen's $f$=0.79; 95% CI, (7.69, 9.38) | $F_{5,160}$=2.50; $P$=0.033; Cohen's $f$=0.58; 95% CI, (2.04, 2.96) | $F_{5,160}$=10.92; $P$<0.001; Cohen's $f$=0.83; 95% CI, (9.96, 11.87) |
| $t$-test between the Global North and Global South | | $t_{164}$=2.14; $P$=0.034; Cohen's $d$=0.36; 95% CI, (0.01, 0.17) | $t_{164}$=5.43; $P$<0.001; Cohen's $d$=0.84; 95% CI, (0.13, 0.32) | $t_{164}$=2.99; $P$=0.003; Cohen's $d$=0.50; 95% CI, (0.05, 0.22) | $t_{164}$=−6.23; $P$<0.001; Cohen's $d$=−1.05; 95% CI, (−0.15, −0.08) | $t_{164}$=−3.02; $P$=0.003; Cohen's $d$=−0.47; 95% CI, (−0.11, −0.02) | $t_{164}$=−5.60; $P$<0.001; Cohen's $d$=−0.97; 95% CI, (−0.15, −0.08) |

Values are presented as mean±standard deviation. We conducted two-sided one-way analysis of variance (ANOVA) tests to examine differences in infrastructure access or access inequality across the three dimensions (economic, social and environmental) and across regions (Europe, Asia, North America, South America, Oceania and Africa), and a two-sided $t$-test to compare infrastructure access or access inequality between the Global North and Global South countries.

## Results

We computed the normalized human access to economic, social and environmental infrastructures at both country and county levels using a spatially population-weighted exposure model (Methods). The results revealed significant differences in infrastructure access across regions and infrastructure types (Fig. 2 and Table 1). At the country level, the mean access value for economic infrastructure was 0.39 ± 0.25, followed by environmental (0.35 ± 0.27) and social (0.29 ± 0.27) infrastructures ($F_{2,163}$ = 6.08; $P$ = 0.002; Cohen's $f$ = 0.87; 95% confidence interval (CI), (5.71, 6.46)). Regions such as Europe, Asia, North America and South America displayed higher access to economic and social infrastructures, while countries in Oceania, South America and Europe showed higher access to environmental infrastructure (economic: $F_{5,160}$ = 7.89; $P$ < 0.001; Cohen's $f$ = 0.78; 95% CI, (7.08, 8.71); social: $F_{5,160}$ = 6.40; $P$ < 0.001; Cohen's $f$ = 0.75; 95% CI, (5.66, 7.13); environmental: $F_{5,160}$ = 16.11; $P$ < 0.001; Cohen's $f$ = 0.87; 95% CI, (14.95, 17.27)). Africa had the lowest access levels for economic (0.24 ± 0.18) and social (0.13 ± 0.11) infrastructures, whereas Asian countries recorded the lowest access to environmental infrastructure (0.21 ± 0.21). Generally, Global North countries experienced greater levels of infrastructure access, with mean values of economic, social and environmental infrastructure access being 1.25, 2.00 and 1.43 times higher than those in the Global South (economic: $t_{164}$ = 2.14; $P$ = 0.034; Cohen's $d$ = 0.36; 95% CI, (0.01, 0.17); social: $t_{164}$ = 5.43; $P$ < 0.001; Cohen's $d$ = 0.84; 95% CI, (0.13, 0.32); environmental: $t_{164}$ = 2.99; $P$ = 0.003; Cohen's $d$ = 0.50; 95% CI, (0.05, 0.22)). This contrasting difference in infrastructure access was also evident at the county level, where the average values of economic, social and environmental infrastructure access were 1.41, 2.63 and 1.22 times higher in the Global North than in the Global South (Supplementary Fig. 2 and Supplementary Table 1).

By adopting the 25th and 75th percentile values as thresholds, we classified each dimension of economic, social and environmental infrastructure access as high (H), medium (M) or low (L), resulting in 27 distinct categories of combinations (Fig. 3). These categories were then grouped into three general classes (Classes I, II and III), representing varying levels of infrastructure access and disparities across the three dimensions (Methods). For example, 'H-M-L' indicates high, medium and low levels for economic, social and environmental infrastructure access, respectively. The results revealed that the most prevalent categories were 'M-M-M' (26 countries), 'H-M-M' (13 countries) and 'H-H-M' (12 countries). Five countries fell into the 'H-H-H' category, including Australia, Canada, Chile, Peru and Portugal, while nine African countries were categorized as 'L-L-L', including Burkina Faso, Central African Republic, Chad, Djibouti, Guinea, Mauritania, Niger, South Sudan and Sierra Leone (Fig. 3a). We also found several countries categorized as 'H-H-L' or 'H-M-L', such as China and India, reflecting relatively high infrastructure access in the socio-economic dimensions but lower access in environmental infrastructure. In general, the majority of countries in the Global North (45 out of 54) were categorized as Class I, indicating that their infrastructure access levels exceeded the average, with relatively small differences among the economic, social and environmental dimensions (a win–win scenario). In contrast, Global South countries presented a more varied picture. While 38 of them (34%) belonged to Class I, a considerable portion fell into Classes II (26%) and III (40%), indicating moderate to below-average infrastructure access levels and considerable disparities among the three dimensions. Similar patterns were also observed at the county scale, with infrastructure access and equity levels in the Global North generally outperforming those in the Global South (Fig. 3b).

We further evaluated the level of inequality in human access to infrastructure at the country scale using the Gini coefficient. The results revealed notable differences in infrastructure access inequality among different types of infrastructure and regions (Fig. 4 and Table 1). Social infrastructure exhibited the highest level of access inequality (mean Gini of 0.83 ± 0.12), followed by economic (mean Gini of 0.55 ± 0.12) and environmental (mean Gini of 0.35 ± 0.14) infrastructures ($F_{2,163}$ = 581.82; $P$ < 0.001; Cohen's $f$ = 1.00; 95% CI, (578.16, 585.49)). Regarding geographical disparities, individuals in Europe and Oceania had more equitable access to infrastructure (economic: $F_{5,160}$ = 8.53; $P$ < 0.001; Cohen's $f$ = 0.79; 95% CI, (7.69, 9.38); social: $F_{5,160}$ = 2.50; $P$ = 0.033;

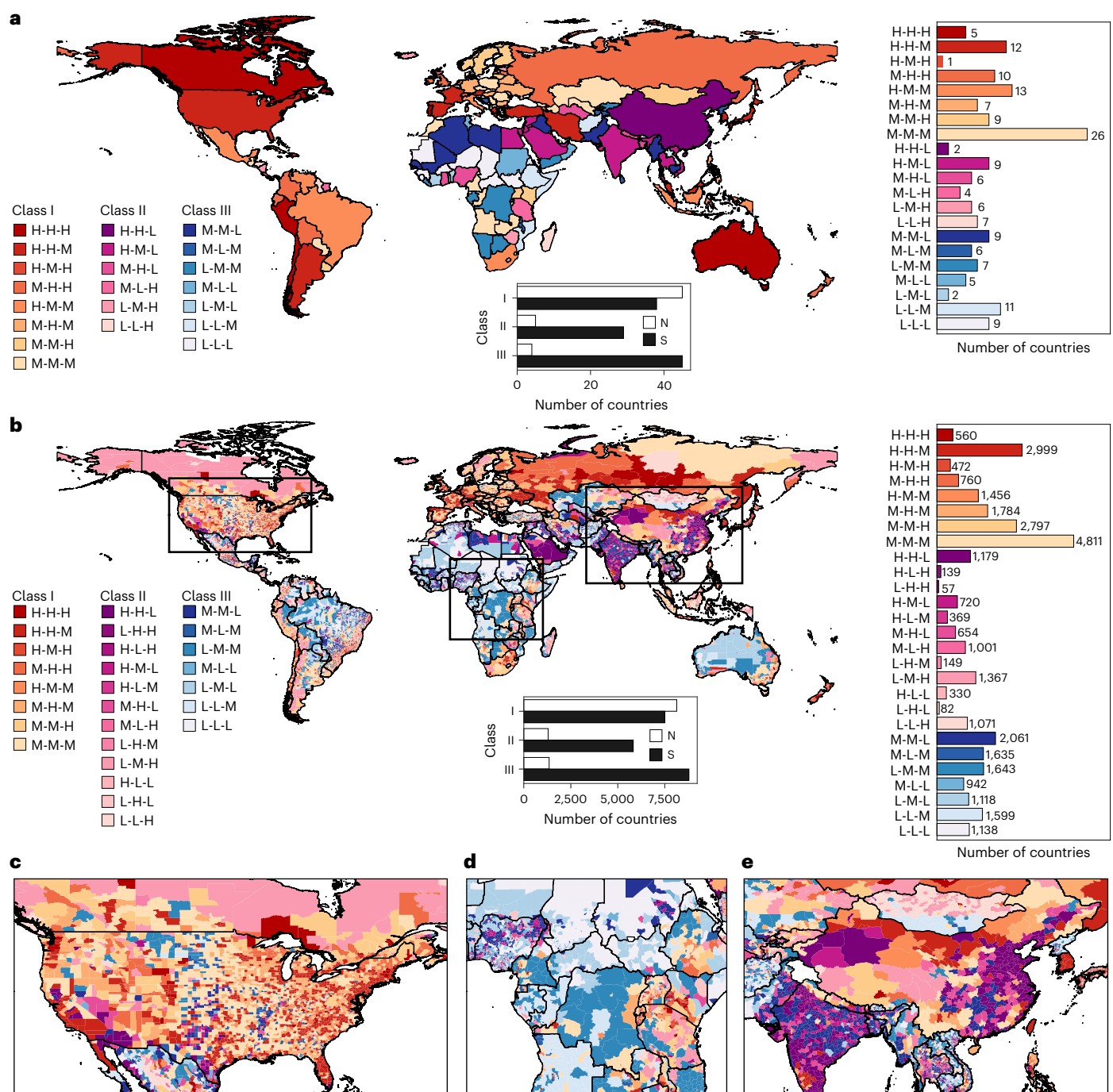

**Fig. 3 | Composite maps of overall infrastructure access levels. a,b,** Access levels at the country (**a**) and county (**b**) scales. We identified 27 unique categories classified into three general classes (I, II and III) representing different levels of infrastructure access and disparities across the three dimensions. For example, 'H-M-L' indicates high, medium and low levels for economic, social and environmental infrastructure access, respectively. The bar charts below the maps illustrate the count of countries or counties within each general class in the Global North (N) and Global South (S). The bar charts on the right show the counts of countries and counties for each category. **c**–**e**, Zoomed composite maps of county-level infrastructure access in North America, Africa and Asia.

Cohen's $f$ = 0.58; 95% CI, (2.04, 2.96); environmental: $F_{5,160}$ = 10.92; $P$ < 0.001; Cohen's $f$ = 0.83; 95% CI, (9.96, 11.87)). Africa experienced extreme economic and social infrastructure access inequality, with mean Gini of 0.61 ± 0.12 and 0.87 ± 0.09, respectively, whereas Asia displayed the highest level of inequality in environmental infrastructure access (mean Gini of 0.43 ± 0.14). In addition to infrastructure access, we observed a difference in infrastructure access inequality between countries of the Global North and Global South (economic: $t_{164}$ = −6.23;

$P$ < 0.001; Cohen's $d$ = −1.05; 95% CI, (−0.15, −0.08); social: $t_{164}$ = −3.02; $P$ = 0.003; Cohen's $d$ = −0.47; 95% CI, (−0.11, −0.02); environmental: $t_{164}$ = −5.60; $P$ < 0.001; Cohen's $d$ = −0.97; 95% CI, (−0.15, −0.08)). The mean values of economic, social and environmental infrastructure access inequality in the Global South were 1.23, 1.09 and 1.44 times higher than those in the Global North. Moreover, by using the inequality index (Inq)[26,51] as an auxiliary measure, we identified similar patterns of spatial inequality in infrastructure access across countries

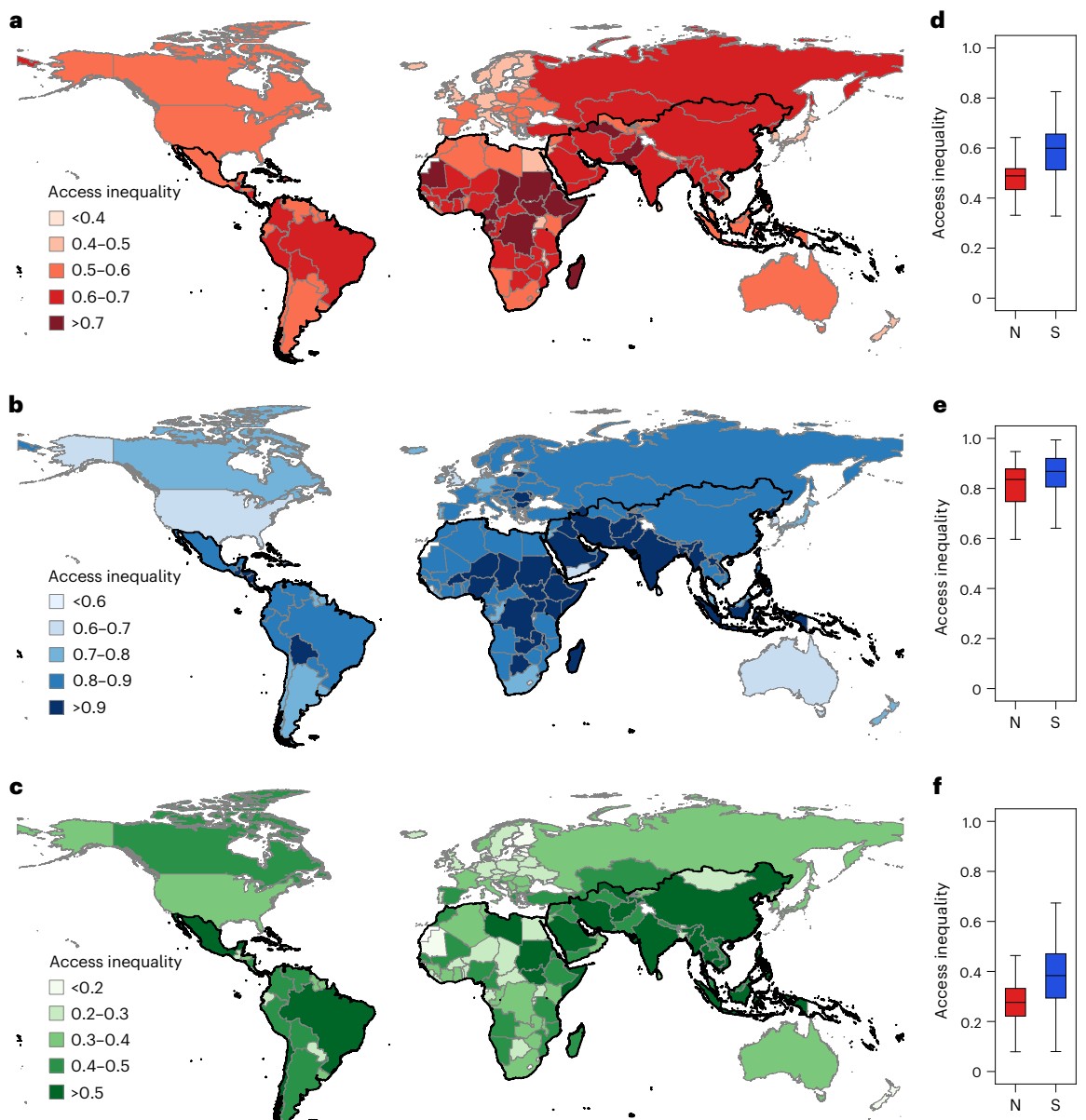

**Fig. 4 | Country-level infrastructure access inequalities measured by the Gini coefficient. a**–**c**, Global maps of economic (**a**), social (**b**) and environmental (**c**) infrastructure access inequalities. Black boundaries indicate countries in the Global South. **d**–**f**, Box plots of economic (**d**), social (**e**) and environmental (**f**) infrastructure access inequalities in countries in the Global North (N) and Global South (S). The box plots display the distribution of the data, with the median (50th percentile) at the centre, the interquartile range (25th to 75th percentiles) as the box, and whiskers extending to the maximum and minimum values within

1.5 times the interquartile range. The sample sizes were 54 countries for the Global North and 112 countries for the Global South. No replicates were carried out, and no adjustments for multiple comparisons were applied. Statistical significance was determined using a two-sided $t$-test for comparisons between the Global North and Global South (economic: $t_{164} = -6.23$; $P < 0.001$; Cohen's $d = -1.05$; 95% CI, $(-0.15, -0.08)$; social: $t_{164} = -3.02$; $P = 0.003$; Cohen's $d = -0.47$; 95% CI, $(-0.11, -0.02)$; environmental: $t_{164} = -5.60$; $P < 0.001$; Cohen's $d = -0.97$; 95% CI, $(-0.15, -0.08)$).

(Supplementary Fig. 3 and Supplementary Table 2). These findings highlight substantial disparities in fundamental infrastructure access and inequality between the Global North and South.

We examined the relationship between infrastructure access, infrastructure access inequality and health-adjusted life expectancy (HALE) using correlational analyses. Both economic and social infrastructure access showed a positive relationship with life expectancy, with Global South countries having steeper regression slopes (Fig. 5a,b). However, increased levels of economic and social infrastructure access inequality were associated with reduced HALE (Fig. 5d,e). No significant relationship was observed between environmental infrastructure access factors and HALE (Fig. 5c). Regarding the association between

environmental infrastructure access inequality and HALE, we found contrasting patterns in countries of the Global North and South, where HALE decreased as inequality levels rose in the Global North, a stark contrast from the Global South (Fig. 5f).

To delve further into the health outcomes associated with socio-economic infrastructure access and inequalities, we employed a linear mixed-effects model for countries in the Global North and Global South, while accounting for potential confounding factors such as gross domestic product (GDP) and population (Methods). The results indicate that increased human access to economic infrastructure (ExpEco) was positively associated with higher HALE (Model I in Table 2). In contrast, economic infrastructure access inequality (GiniEco) showed

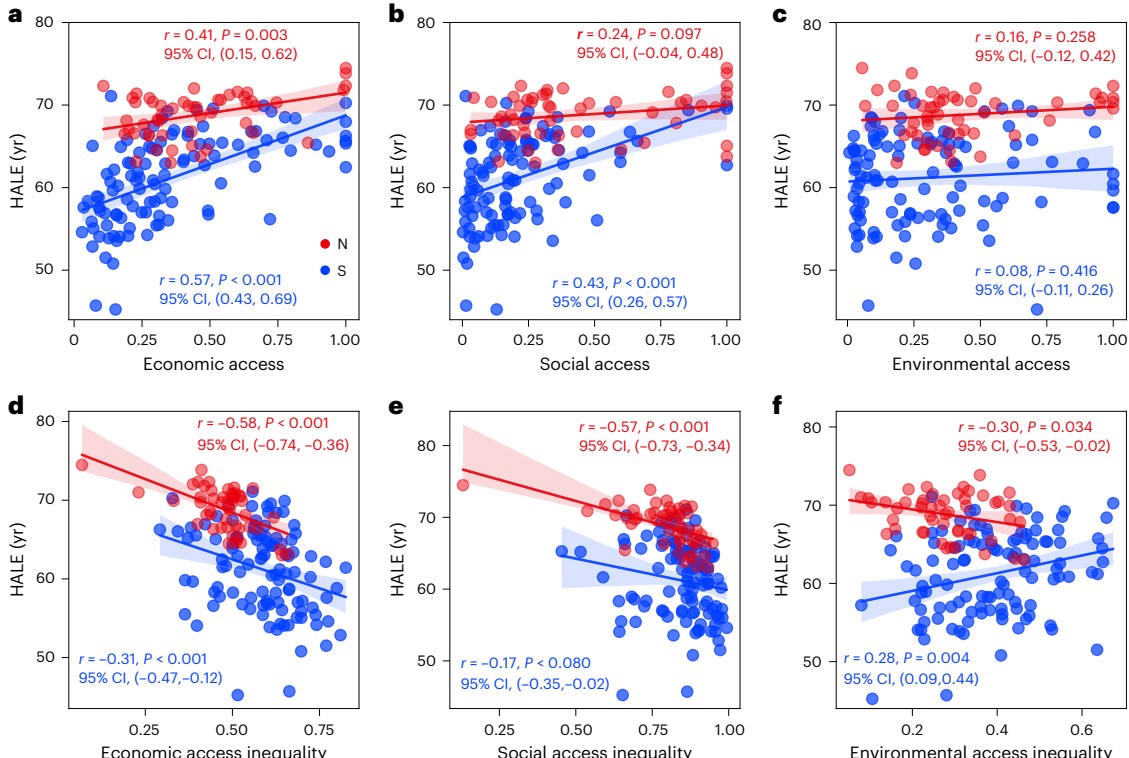

**Fig. 5 | Scatter plots between infrastructure factors and health outcomes in countries in the Global North (N) and Global South (S). a–c**, Associations between infrastructure access and health-adjusted life expectancy (HALE) across economic (**a**), social (**b**) and environmental (**c**) dimensions. **d–f**, Associations between infrastructure access inequality and HALE across economic (**d**), social (**e**) and environmental (**f**) dimensions. The sample sizes were 51 countries for the Global North (red) and 109 countries for the Global South (blue). Ordinary least squares linear regression models were conducted with two-sided hypothesis tests and no multiple comparison adjustments. The shaded areas denote the 95% CIs from the linear regression models, and the central lines represent the predicted values. Pearson correlation coefficients (*r*), their 95% CIs and significance levels (*P*) are reported in each panel.

a negative relationship with HALE (Model II in Table 2). When we integrated all variables into a single linear mixed-effects model (Model III), GiniEco emerged as one of the strongest predictors, with a regression coefficient of −9.95 ($z = −2.96$; $P = 0.003$; 95% CI, (−16.54, −3.36)). This finding suggests that a 10% increase in economic infrastructure access inequality was linked to a one-year reduction in life expectancy. Yet in these models, social infrastructure variables (ExpSoc and GiniSoc) did not show statistically significant correlations with HALE at the $P < 0.05$ level. Additionally, by taking disability-adjusted life years (DALYs) as the response variable in Models I–III, we observed a similar relationship between ExpEco/GiniEco and DALYs (see Supplementary Table 3 for model summaries); that is, lower access levels and higher access inequality levels of economic infrastructure corresponded to a higher disease burden.

## Discussion

Infrastructure is a cornerstone of societal development and well-being, yet the nature of global infrastructure access inequality and its association with human health remains unclear. Our findings address this gap by exposing contrasting disparities in infrastructure access between the Global North and South. Despite hosting 85% of the global population, Global South countries have 50–80% of the infrastructure access but experience 9–44% higher inequalities compared with Global North countries. In particular, African and Asian countries stand out with the lowest levels of infrastructure access and equality, which highlights a severe deficiency in local infrastructure conditions. Although the Global North generally demonstrates higher access and lower inequality in infrastructure than the Global South, this by no means suggests that infrastructure development within these countries is uniform. Even in the developed world, we

find a considerable number of countries with severe inequalities in infrastructure access, particularly in the social dimension, where 42 of the 54 Global North countries have a Gini value larger than 0.7. Our composite maps also highlight distinct regional disparities within Global North countries. For example, in Class I countries such as Canada and the United States, 16% (541 out of 3,402) of their counties are classified under Class II or III (Fig. 3c). Conversely, Class II or III countries, such as those in Africa and Asia, can encompass regions classified as Class I (Fig. 3d,e).

Another contribution of this study lies in the joint assessment of economic–social–environmental infrastructure, which offers an enriched understanding of global infrastructure access and inequality. By categorizing infrastructure access into high, medium and low for each dimension, we identify 27 unique categories of combination, which group into three general classes representing varying levels and disparities in economic, social and environmental infrastructure access. The advantages of our framework lie in its ability to elucidate these levels and disparities across different dimensions, thereby assisting policymakers in more effectively addressing diverse infrastructure challenges[52,53]. For instance, patterns such as 'H-H-L', 'H-M-L' or 'M-H-L' demonstrate well-developed socio-economic infrastructure but often neglected environmental issues in some countries and regions. Conversely, we observe an 'L-L-H' pattern in countries such as Fiji, Guyana, Honduras, Madagascar, Swaziland, Timor-Leste and Vanuatu. These countries benefit from relatively well-developed environmental infrastructure, primarily due to lower exposure to heat and air pollution and greater access to green space (Supplementary Fig. 4); however, they face substantial deficits in economic and social infrastructure, making the promotion of socio-economic development a primary focus for the future. These patterns provide a comprehensive view of

**Table 2 | Summary of linear mixed-effects regression models with health-adjusted life expectancy (HALE) as the response variable**

| | Coefficient | s.e. | z | P>\|z\| | 95% CI (0.025, 0.975) | |
|---|---|---|---|---|---|---|
| Model I ($R^2$=0.60; Levene's test: $F$=11.86, $P$<0.001) | | | | | | |
| ExpEco | 5.85 | 1.54 | 3.81 | <0.001 | 2.84 | 8.86 |
| ExpSoc | 1.79 | 1.31 | 1.37 | 0.171 | −0.77 | 4.36 |
| LnPop | −2.18 | 0.27 | −8.07 | <0.001 | −2.71 | −1.65 |
| LnGDP | 1.89 | 0.31 | 6.11 | <0.001 | 1.29 | 2.50 |
| const | 67.06 | 3.39 | 19.79 | <0.001 | 60.42 | 73.71 |
| Model II ($R^2$=0.65; Levene's test: $F$=10.98, $P$<0.001) | | | | | | |
| GiniEco | −11.69 | 3.24 | −3.61 | <0.001 | −18.04 | −5.33 |
| GiniSoc | −0.54 | 3.04 | −0.18 | 0.859 | −6.50 | 5.42 |
| LnPop | −2.14 | 0.28 | −7.75 | <0.001 | −2.68 | −1.60 |
| LnGDP | 2.61 | 0.24 | 10.75 | <0.001 | 2.14 | 3.09 |
| const | 64.50 | 3.07 | 21.04 | <0.001 | 58.50 | 70.51 |
| Model III ($R^2$=0.66; Levene's test: $F$=12.79, $P$<0.001) | | | | | | |
| ExpEco | 3.84 | 1.82 | 2.11 | 0.035 | 0.28 | 7.41 |
| ExpSoc | 2.23 | 1.32 | 1.69 | 0.091 | −0.36 | 4.82 |
| GiniEco | −9.95 | 3.36 | −2.96 | 0.003 | −16.54 | −3.36 |
| GiniSoc | 3.03 | 3.22 | 0.94 | 0.346 | −3.28 | 9.35 |
| LnPop | −2.07 | 0.27 | −7.64 | <0.001 | −2.61 | −1.54 |
| LnGDP | 2.01 | 0.32 | 6.34 | <0.001 | 1.39 | 2.64 |
| const | 66.86 | 3.21 | 20.82 | <0.001 | 60.56 | 73.15 |

The sample sizes were 51 countries for the Global North and 109 countries for the Global South. Normality was tested using the Kolmogorov–Smirnov test ($D$=0.096, $P$=0.101), indicating that log transformation was not necessary for the response variable. Homogeneity of variances was assessed using Levene's test on residuals, and robust standard errors were applied using the sandwich estimator when $P$<0.05. An asterisk denotes a significance level of $P$<0.05. All $P$ values are from two-sided tests, with no adjustments for multiple comparisons.

global infrastructure access, pinpointing areas where human access to specific types of infrastructure is most fragile, inequitable and urgently in need of aid.

Moreover, the population-weighted exposure model employed in this study takes into account the spatial distribution of both population and infrastructure, enabling a more accurate representation of human–infrastructure interactions (see Supplementary Information section 2 for further illustrations in Australia and Burkina Faso). Traditional measurements of infrastructure access often assume a static population distribution across space and time[10,12], which can result in biases when assessing realistic human access to infrastructure. To illustrate this, we further compute the country-level differences between population-weighted access estimates and original values for economic, social and environmental infrastructure categories. The results suggest that traditional intuitive measurement methods could potentially overestimate or underestimate the real situation (Supplementary Fig. 5). In most African, Asian and European countries, population-weighted access values for economic infrastructure are generally below the original values, indicating that a large portion of the population resides in areas that are far from business facilities (Supplementary Fig. 5a). For social and environmental infrastructure, the original values tend to underestimate their actual levels, albeit to a lesser extent in Africa and Asia than in other regions (Supplementary Fig. 5b,c). Our population-weighted infrastructure access framework reinforces the importance of assessing the adequacy of infrastructure provisions through the lens of human–infrastructure supply–demand relationships.

Our analysis also sheds light on the linkages between infrastructure and human health, underscoring that equitable access to infrastructure, particularly in the economic dimension, is fundamental for enhancing human health and well-being. At the country scale, we find a positive relationship between infrastructure access and HALE and a negative relationship between infrastructure access inequality and HALE (Fig. 5). In the United States, similar patterns appear at the county scale, where greater infrastructure access values correspond to higher life expectancy and lower disease mortality rates (Supplementary Fig. 6). These findings are generally consistent with previous studies on the relationship between individual types of infrastructure and health outcomes across regions[39–45]. Our mixed-effects regression modelling further advances current knowledge by offering a quantitative comparison of health disparities across different infrastructure types and between levels of access and inequality. The coefficient for economic infrastructure inequality (GiniEco, Table 2) underscores the key role of infrastructure inequality—a factor largely underexplored in the literature.

In practice, these findings offer policy implications for equitable and sustainable infrastructure development to promote human health. First, our results emphasize the need for targeted policies and interventions to bridge infrastructure gaps, especially for countries in the Global South. According to the United Nations, the world's population will continue to increase in the coming decades, projected to reach 9.7 billion by 2050 and 10.4 billion by 2100[54]. Much of this growth will occur in the developing world, posing potential threats to many Global South countries, as they strive to meet the rising demands for essential infrastructure services such as water, sanitation, education and health. Our maps and findings serve as a reference for governments and policymakers to identify regions with low infrastructure access and high inequality, which will facilitate more strategic investments that ensure equitable access to essential services and opportunities. By prioritizing infrastructure development in these identified under-resourced areas, policymakers can promote inclusive growth that supports overall economic and social advancement.

Second, our findings underline formidable challenges in accomplishing SDG 9 and beyond, as infrastructure inequality remains a barrier to sustainable and equitable development. While infrastructure investment is essential, economic growth alone cannot resolve infrastructure disparities; in some cases, it may even exacerbate regional disparities due to policies or investments that favour specific neighbourhoods over others[55,56]. At the national level, such targeted investments and policies can further widen disparities across regions and cities[57–59], and if population growth outpaces infrastructure construction, disparities in infrastructure access may worsen[26]. To effectively address these challenges, policymakers need to prioritize equitable infrastructure allocation on the basis of a spatial assessment of infrastructure needs across multiple dimensions, regions and human–infrastructure interaction settings. This can involve implementing redistributive policies that channel resources to historically underserved regions and enforcing guidelines to prevent infrastructure development projects from favouring particular neighbourhoods at the expense of others. Policymakers should also account for real-time population dynamics to ensure that infrastructure expansions can accommodate rising demand, especially in regions where rapid population increases may otherwise outstrip infrastructure improvements.

Lastly, our analysis emphasizes the importance of reducing infrastructure disparities to improve health outcomes. While the provision of infrastructure services is fundamental to human health, addressing disparities in access, especially in regions with low overall infrastructure access, is even more vital. Our findings reveal that even small increases in inequality, particularly in the economic dimension, can compromise health outcomes. This necessitates targeted efforts to lessen inequalities in key infrastructure components such

as transportation, telecommunications and housing. By implementing policies that secure universal and inclusive infrastructure access, policymakers can foster healthier and more resilient communities, further contributing to the global goal of improved well-being and quality of life.

We also acknowledge several levels of limitations that suggest directions for future investigations. First, while we examine infrastructure from economic, social and environmental perspectives, we note that infrastructure classifications are diverse, with no consistent standards in the literature[2,60]. In some cases, categories may overlap; for instance, water and waste infrastructure can be considered economic infrastructure[61,62]. Our focus, however, is on developing a flexible comparative framework, which can accommodate a wide range of critical infrastructure systems and effectively reveal access inequalities across different types. Future research could adapt this model to fit various infrastructure classifications based on specific application needs. Second, the population-weighted exposure model used in this study assumes static population distributions across space and time, focusing on physical access to nearby infrastructure rather than accounting for spatiotemporal interactions between infrastructure and mobile individuals. However, as people move throughout their daily lives, they may benefit from distant infrastructure (for example, hospitals and schools) or experience negative impacts from nearby infrastructure (for example, noise and air pollution from highways). Moreover, spatial proximity does not always guarantee higher accessibility due to spatial segregation, such as gated communities or restricted-access roads and facilities[63–66]. To capture these dynamics more realistically, future studies should consider diverse types of infrastructure accessibility beyond mere spatial proximity and incorporate human mobility data to assess human-centric exposure through a spatiotemporally explicit interaction framework. Third, although this research reveals health disparities associated with various infrastructure types, it does not consider the effect of interdependencies among these types[67]. For example, many critical infrastructure sectors cannot operate if energy infrastructure is not functioning[68]. This is further supported by our correlation analysis, where access to transportation, energy and health infrastructure exhibits higher correlations than access to other types (Supplementary Fig. 7). In the next stage, more work is needed to understand how these interdependencies collectively impact human health. Finally, there are data gaps and limitations associated with the infrastructure dataset used in this study. On the one hand, as the critical infrastructure dataset is derived from a voluntary data source, OpenStreetMap (OSM)[69], it may contain missing data, particularly in less-developed areas, which limits the representation of certain countries or regions. On the other hand, our inequality analysis is conducted at the national scale, constrained by the dataset's relatively coarse spatial resolution (0.1° × 0.1°). This approach does not capture intra-country heterogeneity in inequality levels, as disparities in infrastructure provision can exist even within developed countries with low overall inequality. Moving forward, we plan to integrate multi-source data, such as high-resolution satellite imagery and human mobility data, to analyse infrastructure exposure inequalities at finer spatial scales and identify vulnerable hotspots requiring targeted policy interventions and initiatives.

## Methods

### Research design and data

Supplementary Fig. 1 provides a detailed flow chart of the study, outlining the research design. First, we combined multi-source geospatial datasets to generate a comprehensive dataset of global infrastructure distributions across the economic, social and environmental dimensions. Second, we applied population-weighted assessment to gridded population data and infrastructure mapping results, quantifying spatial differences in human access to infrastructure distributions at the country and county levels, as well as between Global North and Global South countries. Third, we evaluated global inequality in infrastructure access by comparing the cumulative population and infrastructure distributions in each administrative region of each country. Finally, we performed statistical analysis to explore the relationship between infrastructure access/inequality and human health. All the data sources used in this study are publicly available (Supplementary Table 4). Additional details on data acquisition and processing are provided in Supplementary Information section 3.

### Infrastructure mapping

On the basis of the conceptual model (Fig. 1), we built three globally harmonized and consistent maps of economic, social and environmental infrastructures for 2020 at 0.1° × 0.1° spatial resolution using a range of open data sources, including critical infrastructure, land cover, air pollution and re-analysis climate data. To begin with, we reclassified the original 39 infrastructure types in the critical infrastructure dataset[69] into economic, social and environmental categories. For economic infrastructure, we considered telecommunication, energy and transport systems. For social infrastructure, we considered health and education systems. For environmental infrastructure, we incorporated water and waste systems.

In the original critical infrastructure dataset, the raster value represents the total amount of infrastructure (for example, the number of power poles) within a given grid cell[69]. Statistics show that some infrastructure types, such as power poles and tertiary roads, have significantly higher quantities than others (Supplementary Table 5). To enable cross-type comparisons, we normalized each infrastructure type layer within each critical infrastructure system to a 0–1 scale by dividing by the maximum grid value for that type (Supplementary Fig. 8). We then aggregated the normalized infrastructure type layers to the initial economic, social and environmental infrastructure layers by equal weighting. For economic and social infrastructures, we later calibrated the maps with the night-time light (NTL) data (as described in the subsequent section).

In terms of environmental infrastructure, we integrated the normalized water and waste infrastructure layer with data on green space, air pollution and thermal comfort. Generally, a superior condition of environmental infrastructure is indicated by an increased presence of water and waste facilities, higher green space coverages, lower $PM_{2.5}$ concentrations and shorter heat durations. The equation for the calculation of environmental infrastructure (Env) can therefore be expressed as:

$$Env = 0.5 \times CI_{env} + 0.5 \times \frac{e^{Green}}{\ln(e^{Air} \times e^{Heat})} \quad (1)$$

where $CI_{env}$ represents the initial environmental infrastructure layer, reclassified from the critical infrastructure dataset as the mean of the normalized water and waste infrastructure layers. Green is the green space layer extracted from the WorldCover data, Air is the normalized $PM_{2.5}$ layer calculated on the basis of the Goddard Earth Observing System Composition Forecast data and Heat is the normalized heat duration layer derived from the ERA5 climate reanalysis data. All green space, $PM_{2.5}$ and heat duration layers were resampled to 0.1° × 0.1° before processing.

### Socio-economic infrastructure data calibration

The CI dataset used in this study was derived from OSM, a geographic database updated and maintained by a community of volunteers via open collaboration[70]. However, concerns have been raised regarding the quality of OSM data given its collaborative nature[71,72]. A recent study assessing the accuracy of OSM land-cover/land-use data in 168 countries worldwide found that data completeness in 129 countries was less than 40%[73]. This suggests that in some regions, particularly in developing regions, the spatial coverage of OSM data remains incomplete.

To mitigate potential biases stemming from the limitations of OSM data, we integrated VIIRS NTL data as a surrogate to calibrate the economic and social infrastructure layers derived from the CI dataset. Previous studies have demonstrated the role of night-time lights in estimating various socio-economic activities including economic growth[74,75], electricity consumption[76,77] and population distribution[78,79]. In practice, we selected the continental United States as the calibration site due to its relatively comprehensive coverage of OSM data[80]. Recognizing that most socio-economic activities are concentrated in urban areas, we employed buffer areas of global urban boundary (GUB) data within the continental United States to exclude pixels located outside of urban regions. We conducted linear regression analyses between NTL and infrastructure values at a spatial scale of $0.1° × 0.1°$, using various buffer sizes ranging from 0 to 25 km. The results revealed strong positive relationships between logarithmic values of night-time lights and economic infrastructure (Supplementary Fig. 9), with the most robust linear relationship observed within the 5-km buffer area of GUB ($r = 0.71$, $P < 0.001$; Supplementary Fig. 9b). Consequently, we used coefficients of the 5-km model to estimate the final economic and social infrastructure distributions on a global scale, which can be expressed as follows:

$$\ln(I) = 1.58 × \ln(I') + 5.03 \tag{2}$$

where $I$ denotes the calibrated economic (social) infrastructure value within each $0.1° × 0.1°$ grid cell, while $I'$ is the initially normalized economic (social) infrastructure value derived from the CI dataset.

To validate the accuracy and reliability of the calibrated infrastructure maps, we conducted a comparison between the economic and social infrastructure layers and national-level data on GDP and the Human Development Index (HDI), respectively. GDP is a widely recognized and dependable indicator for measuring a region's economic performance, while the HDI represents the overall human development achievement across various dimensions, including health, education and standard of living[81]. Prior studies have demonstrated the positive impact of social infrastructure on human development[82–84]. Our analysis supported the existing evidence by revealing a statistically significant linear relationship between the sum of social infrastructure and HDI across countries ($r = 0.89$, $P < 0.001$; Supplementary Fig. 10b). We also observed a similar relationship between the sum of economic infrastructure and GDP at the country level ($r = 0.90$, $P < 0.001$; Supplementary Fig. 10a).

### Human access to infrastructure

We employed the population-weighted exposure model—a bottom-up assessment to quantify the level of human access to infrastructure. The model has been developed to capture the spatial interaction between the population and various indicators of the surrounding environment, including green space[15,16,85], air pollution[86,87] and thermal comfort[88]. When applying this model to measure infrastructure access, we proportionately assigned higher weights to infrastructures located in areas with larger populations:

$$IE^k = \frac{\sum_{i=1}^{N} P_i × I_i^k}{\sum_{i=1}^{N} P_i} \tag{3}$$

where $P_i$ is the population of the $i$th grid, $I_i^k$ is the infrastructure value of the $i$th grid for type $k$ (that is, economic, social or environmental), $N$ is the total number of grids within the corresponding administrative unit and $IE^k$ is the corresponding population-weighted infrastructure access level for type $k$ (that is, economic, social or environmental). For each type $k$, we normalized $IE^k$ by dividing it by the 95th percentile of $IE^k$, capping any values exceeding this threshold to 1. This resulted in a standardized IE range of 0–1, with higher values indicating greater human access to infrastructure.

### Comprehensive evaluation of infrastructure access

To assess overall infrastructure accessibility and disparities among economic, social and environmental dimensions, we categorized the resulting infrastructure access values as 'high' (H), 'medium' (M) and 'low' (L) for each dimension using the 25th and 75th quantiles as thresholds. Next, we synthesized the three-dimensional classification outcomes into composite maps at both county and country scales, resulting in 27 distinct categories (Fig. 3). Each uppercase letter denotes an infrastructure access level for a specific dimension. For instance, 'H-M-L' denotes high, medium and low levels for economic, social and environmental infrastructure access, respectively. Furthermore, we grouped the 27 categories into three general classes based on their characteristics:

- Class I denotes regions where infrastructure access levels surpass the average, with relatively minor disparities among the three dimensions (none falling into 'low')
- Class II represents regions with moderate overall infrastructure access but notable disparities across the three dimensions (at least one 'high' and one 'low')
- Class III indicates infrastructure access levels below the average, with relatively minor disparities among the three dimensions (none reaching 'high')

### Infrastructure access inequality

We used the widely adopted Gini coefficient[89] (Gini) to assess the inequality in human access to infrastructure for each of the 166 countries in the Global North and Global South. The Gini coefficient is a statistical measure of dispersion that compares the cumulative proportions of the population against the cumulative proportions of infrastructures they have access to[90]. It provides a numerical value ranging from 0 to 1, where larger values indicate a higher level of inequality in infrastructure access, and vice versa. Additionally, we introduced the inequality index[26,51] (Inq) as a complementary measure of the spatial inequality in infrastructure access. The inequality index also ranges from 0 to 1, with 0 representing absolute equality and 1 representing absolute inequality[12]. More details on the calculation of Gini and Inq are provided in Supplementary Information section 4.

### Associations between infrastructure factors and human health

We conducted correlation analyses between infrastructure access or infrastructure access inequality and health outcomes (that is, HALE and DALYs) in 2020 at the country level. Both economic and social infrastructure access had a positive relationship with life expectancy (Fig. 5a,b), while greater inequality of economic and social infrastructure access was associated with lower HALE (Fig. 5d,e). Our analyses also highlighted the difference in the relationships between countries in the Global North and Global South, as outlined by red and blue in Fig. 5. However, we found no significant relationship between environmental infrastructure access factors and HALE (Fig. 5c).

To further examine the association between infrastructure access, inequality and human health, we performed linear mixed-effects regression modelling[91] between socio-economic infrastructure factors and health outcomes for countries in the Global North and Global South. The first model (Model I) included only infrastructure access variables. The second model (Model II) included only infrastructure access inequality variables. The third model (Model III) included all the variables of infrastructure access and inequality included in Models I and II. In addition, we added measures of population and GDP to control for the unobserved entity characteristics among countries. The mathematical expressions for Models I–III are outlined in equations (4)–(6), respectively:

$$
\begin{aligned}
\text{Health}_{j,g} = {} & \beta_0 + \gamma_g + \beta_1 \text{LnPop}_{j,g} + \beta_2 \text{LnGDP}_{j,g} \\
& + \beta_3 \text{ExpEco}_{j,g} + \beta_4 \text{ExpSoc}_{j,g} + \varepsilon_{i,g}
\end{aligned}
\tag{4}
$$

$$\text{Health}_{j,g} = \beta_0 + \gamma_g + \beta_1 \text{LnPop}_{j,g} + \beta_2 \text{LnGDP}_{j,g}$$
$$+ \beta_3 \text{GiniEco}_{j,g} + \beta_4 \text{GiniSoc}_{j,g} + \varepsilon_{i,g} \qquad (5)$$

$$\text{Health}_{j,g} = \beta_0 + \gamma_g + \beta_1 \text{LnPop}_{j,g} + \beta_2 \text{LnGDP}_{j,g} + \beta_3 \text{ExpEco}_{j,g}$$
$$+ \beta_4 \text{ExpSoc}_{j,g} + \beta_5 \text{GiniEco}_{j,g} + \beta_6 \text{GiniSoc}_{j,g} + \varepsilon_{j,g} \qquad (6)$$

where $\text{Health}_{i,g}$ denotes health outcomes for country $j$ in group $g$ (that is, Global North or Global South). The covariate variables included the logarithmic value of population (LnPop), the logarithmic value of GDP (LnGDP), economic infrastructure access (ExpEco), social infrastructure access (ExpSoc), economic infrastructure access inequality (GiniEco) and social infrastructure access inequality (GiniSoc). A mixed-effects approach was used, with a random intercept term ($\gamma_g$) accounting for variations in the baseline health conditions across countries in the Global North and Global South. $\beta_0$ is the average model intercept, $\beta_{1-6}$ are the regression coefficients for each covariate variable and $\varepsilon_{j,g}$ is the error term. Model significance was tested at the $P < 0.05$ level. In Table 2, we report mixed-effects regression results taking HALE as the response variable. We also replaced $\text{Health}_{i,g}$ with DALYs in Models I–III to examine the relationship between socio-economic infrastructure factors and disease burdens (see Supplementary Table 3 for model summaries).

Moreover, we applied a machine learning algorithm, the random forest model[92], to build the association between all six covariate variables in Models I–III and health outcomes. Variable importance was quantified by the indicators of the increase in mean square error and the increase of node purity[93]. For each of the response variables HALE and DALYs, we constructed a random forest model with 500 trees and iteratively executed the model 100 times (Supplementary Figs. 11 and 12).

### Sensitivity analyses
We conducted two sensitivity analyses to further validate the robustness of our assessment model. First, we included water and waste infrastructure within the economic and social infrastructure categories to assess how this inclusion affects country-level access values (Supplementary Fig. 13). Second, we quantified the Pearson correlations between various types of infrastructure access (Supplementary Fig. 7) and examined how interactions between infrastructure types relate to human health outcomes (Supplementary Table 6). Further details can be found in Supplementary Information section 5.

### Reporting summary
Further information on research design is available in the Nature Portfolio Reporting Summary linked to this article.

## Data availability
The public datasets used in this study can be accessed via the Google Earth Engine platform (https://earthengine.google.com), specifically including the Harmonized Global Critical Infrastructure dataset (https://gee-community-catalog.org/projects/cisi/), the WorldCover Global Land Cover map for 2020 (https://developers.google.com/earth-engine/datasets/catalog/ESA_WorldCover_v100), NASA's Goddard Earth Observing System Composition Forecast data (https://developers.google.com/earth-engine/datasets/catalog/NASA_GEOS-CF_v1_rpl_tavg1hr), the ERA5-Land Daily Aggregated Climate Reanalysis data (https://developers.google.com/earth-engine/datasets/catalog/ECMWF_ERA5_LAND_DAILY_AGGR) and the WorldPop Global Project Population Data (https://developers.google.com/earth-engine/datasets/catalog/WorldPop_GP_100m_pop). The annual global VIIRS NTL V2 product is from the Earth Observation Group (https://eog-data.mines.edu/products/vnl/#annual_v2). The GUB datasets are from Pengcheng Laboratory (https://data-starcloud.pcl.ac.cn/). The Global Administrative Unit Layers are from the Food and Agriculture

Organization of the United Nations (https://data.apps.fao.org/). The list of Global South countries was obtained from the Organization for Women in Science for the Developing World (https://owsd.net/). Data on GDP are from the World Bank (https://data.worldbank.org). Data on the HDI are from the United Nations Development Programme (https://hdr.undp.org/). Health data on HALE and DALYs are from the Institute for Health Metrics and Evaluation at the University of Washington (https://vizhub.healthdata.org/gbd-results/). The resulting maps of economic, social and environmental infrastructure, along with data on infrastructure access, infrastructure access inequality and health outcomes, are available via figshare at https://figshare.com/projects/Infrastructure_inequality/237854 (ref. 94).

## Code availability
All code used to generate the data and results in this study is available via figshare at https://figshare.com/projects/Infrastructure_inequality/237854 (ref. 94).

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

## Acknowledgements

This study was jointly supported by the National Key Research and Development Program of China (grant nos 2022YFB3903703 to B.X. and B.C. (The University of Hong Kong), 2022YFE0209300 to B.X. and 2024YFF1307000 to B.C. (Beijing Normal University), the National Natural Science Foundation of China (grant nos 42090015 to P.G. and B.X., 42201373 to B.C. (The University of Hong Kong) and 72091511 to B.C. (Beijing Normal University)), the Chinese Academy of Engineering's Strategic Research and Consulting Project (grant no. 2024-XZ-34 to B.X.), Open Research Program of the International Research Center of Big Data for Sustainable Development Goals (grant no. CBAS2022ORP02 to B.X.), the University of Hong Kong HKU-100 Scholars Fund (to B.C. (The University of Hong Kong)), the Seed Fund for Strategic Interdisciplinary Research Scheme Fund and Basic Research, the Research Grants Council of Hong Kong Early Career Scheme and General Research Fund (grant nos HKU27600222 and HKU17601423 to B.C. (The University of Hong Kong)), the NSFC/RGC Joint Research Scheme (grant no. N_HKU722/23 to B.C. (The University of Hong Kong)), the Croucher Foundation (grant no. CAS22902/CAS22HU01 to P.G.), the University of Hong Kong Start-Up Fund for New Professoriate Staff (to J.A.), the Research Grants Council of the Hong Kong Special Administrative Region (grant no. T35/710/20R to C. Lin), the University of Hong Kong Faculty of Business and Economics and Shenzhen Research Institutes (grant no. SZRI2023-CRF-04 to C. Lin), and Cornell Center for Social Sciences (to C. Liao).

## Author contributions

B.C. (The University of Hong Kong) and B.X. conceived the research idea. Y.T., B.C. (The University of Hong Kong) and B.X. designed the study. Y.T. collected the data and performed the main analysis. Y.T. and B.C. (The University of Hong Kong) drafted the manuscript. C. Liao, S.W., J.A., C. Lin, P.G., B.C. (Beijing Normal University) and H.W. contributed to interpreting the results. All authors reviewed and revised the manuscript.

## Competing interests

The authors declare no competing interests.

## Additional information

**Correspondence and requests for materials** should be addressed to Bin Chen, Chuan Liao or Bing Xu.

# Reporting Summary

## Statistics

For all statistical analyses, confirm that the following items are present in the figure legend, table legend, main text, or Methods section.

| n/a | Confirmed | |
|---|---|---|
| ☐ | ☒ | The exact sample size (*n*) for each experimental group/condition, given as a discrete number and unit of measurement |
| ☐ | ☒ | A statement on whether measurements were taken from distinct samples or whether the same sample was measured repeatedly |
| ☐ | ☒ | The statistical test(s) used AND whether they are one- or two-sided *Only common tests should be described solely by name; describe more complex techniques in the Methods section.* |
| ☐ | ☒ | A description of all covariates tested |
| ☐ | ☒ | A description of any assumptions or corrections, such as tests of normality and adjustment for multiple comparisons |
| ☐ | ☒ | A full description of the statistical parameters including central tendency (e.g. means) or other basic estimates (e.g. regression coefficient) AND variation (e.g. standard deviation) or associated estimates of uncertainty (e.g. confidence intervals) |
| ☐ | ☒ | For null hypothesis testing, the test statistic (e.g. $F$, $t$, $r$) with confidence intervals, effect sizes, degrees of freedom and $P$ value noted *Give P values as exact values whenever suitable.* |
| ☒ | ☐ | For Bayesian analysis, information on the choice of priors and Markov chain Monte Carlo settings |
| ☒ | ☐ | For hierarchical and complex designs, identification of the appropriate level for tests and full reporting of outcomes |
| ☐ | ☒ | Estimates of effect sizes (e.g. Cohen's *d*, Pearson's *r*), indicating how they were calculated |

*Our web collection on statistics for biologists contains articles on many of the points above.*

## Software and code

Policy information about availability of computer code

| Data collection | No software was used for data collection. |
|---|---|
| Data analysis | All data processing and analysis were conducted using Google Earth Engine, Python (version 3.11), and ArcMap (version 10.8). |

For manuscripts utilizing custom algorithms or software that are central to the research but not yet described in published literature, software must be made available to editors and reviewers. We strongly encourage code deposition in a community repository (e.g. GitHub). See the Nature Portfolio guidelines for submitting code & software for further information.

## Data

Policy information about availability of data

All manuscripts must include a data availability statement. This statement should provide the following information, where applicable:

- Accession codes, unique identifiers, or web links for publicly available datasets
- A description of any restrictions on data availability
- For clinical datasets or third party data, please ensure that the statement adheres to our policy

Data Availability
A number of public datasets used in this study can be accessed via the Google Earth Engine platform (https://earthengine.google.com), specifically including:
• The Harmonized Global Critical Infrastructure dataset (https://gee-community-catalog.org/projects/cisi/)
• The WorldCover Global Land Cover map for 2020 (https://developers.google.com/earth-engine/datasets/catalog/ESA_WorldCover_v100)

Short

• NASA's Goddard Earth Observing System Composition Forecast (GEOS-CF) data (https://developers.google.com/earth-engine/datasets/catalog/NASA_GEOS-CF_v1_rpl_tavg1hr)
• The ERA5-Land Daily Aggregated Climate Reanalysis data (https://developers.google.com/earth-engine/datasets/catalog/ECMWF_ERA5_LAND_DAILY_AGGR)
• The WorldPop Global Project Population Data (https://developers.google.com/earth-engine/datasets/catalog/WorldPop_GP_100m_pop)
The annual global VIIRS nighttime lights (VNL) V2 product is from the Earth Observation Group (https://eogdata.mines.edu/products/vnl/#annual_v2). The global urban boundary (GUB) datasets are from Pengcheng Laboratory (https://data-starcloud.pcl.ac.cn/). The Global Administrative Unit Layers (GAULs) are from the Food and Agriculture Organization of the United Nations (https://data.apps.fao.org/). The list of Global South countries was obtained from the Organization for Women in Science for the Developing World (https://owsd.net/). Data on gross domestic product (GDP) are from the World Bank (https://data.worldbank.org). Data on Human Development Index (HDI) are from the United Nations Development Programme (https://hdr.undp.org/). Health data on health-adjusted life expectancy (HALE) and disability-adjusted life years (DALYs) are from the Institute for Health Metrics and Evaluation at the University of Washington (https://vizhub.healthdata.org/gbd-results/).
The resulting maps of economic, social, and environmental infrastructure, along with data on infrastructure access, infrastructure access inequality, and health outcomes, have been deposited in an open repository at https://figshare.com/projects/Infrastructure_inequality/237854.

# Research involving human participants, their data, or biological material

Policy information about studies with human participants or human data. See also policy information about sex, gender (identity/presentation), and sexual orientation and race, ethnicity and racism.

| | |
|---|---|
| Reporting on sex and gender | n/a |
| Reporting on race, ethnicity, or other socially relevant groupings | n/a |
| Population characteristics | n/a |
| Recruitment | n/a |
| Ethics oversight | n/a |

Note that full information on the approval of the study protocol must also be provided in the manuscript.

# Field-specific reporting

Please select the one below that is the best fit for your research. If you are not sure, read the appropriate sections before making your selection.

☐ Life sciences ☒ Behavioural & social sciences ☐ Ecological, evolutionary & environmental sciences

For a reference copy of the document with all sections, see nature.com/documents/nr-reporting-summary-flat.pdf

# Behavioural & social sciences study design

All studies must disclose on these points even when the disclosure is negative.

| | |
|---|---|
| Study description | We combined existing critical infrastructure datasets, land cover products, air pollution data, and reanalysis climate data to generate global maps of economic, social, and environmental infrastructure distributions. We employed a population-weighted exposure model to reveal spatial differences in human access to infrastructure and the associated inequality levels between Global North and Global South countries. Additionally, we performed mixed-effects regression analysis to explore the relationship between infrastructure access/inequality and human health. |
| Research sample | We analyzed infrastructure access inequality and the associated health disparities across 166 countries worldwide, including 54 in the Global North and 112 in the Global South. This study sample is representative, covering more than 99% of the global population (~7.77 billion). This broad geographic coverage enables a comprehensive analysis of infrastructure access inequality and associated health disparities across diverse economic and regional contexts. |
| Sampling strategy | We assessed human access to infrastructure at both the country and county levels. Infrastructure access inequalities and regression analyses between infrastructure access/inequality and human health were conducted at the country level. The sample size was not predetermined using a formal statistical method. Instead, only countries with both infrastructure and population distribution were considered, resulting in a final sample of 166 countries. This sample size, representing over 99% of the global population, is sufficient to capture patterns of human access and inequality. |
| Data collection | All the data used in this study were from international organizations, open-source databases and peer-reviewed papers. These data were accessed through direct website downloads or the Google Earth Engine platform (https://earthengine.google.com). |
| Timing | We used the year 2020 as a baseline to elucidate the results and findings. Data were collected between March 2023 and May 2023. |
| Data exclusions | n/a |
| Non-participation | No participants were involved in the study. |

| | Randomization | Randomization was not relevant to this study because our analysis was based on multi-source geospatial datasets rather than a subset of sampled observations. We included all countries with available infrastructure and population distribution data, ensuring broad coverage rather than selecting a random sample. This approach allows for a systematic and representative assessment of infrastructure access and inequality across diverse economic and regional contexts. |
|---|---|---|

# Reporting for specific materials, systems and methods

We require information from authors about some types of materials, experimental systems and methods used in many studies. Here, indicate whether each material, system or method listed is relevant to your study. If you are not sure if a list item applies to your research, read the appropriate section before selecting a response.

## Materials & experimental systems

| n/a | Involved in the study |
|---|---|
| ☒ | ☐ Antibodies |
| ☒ | ☐ Eukaryotic cell lines |
| ☒ | ☐ Palaeontology and archaeology |
| ☒ | ☐ Animals and other organisms |
| ☒ | ☐ Clinical data |
| ☒ | ☐ Dual use research of concern |
| ☒ | ☐ Plants |

## Methods

| n/a | Involved in the study |
|---|---|
| ☒ | ☐ ChIP-seq |
| ☒ | ☐ Flow cytometry |
| ☒ | ☐ MRI-based neuroimaging |

## Plants

| | Seed stocks | *Report on the source of all seed stocks or other plant material used. If applicable, state the seed stock centre and catalogue number. If plant specimens were collected from the field, describe the collection location, date and sampling procedures.* |
|---|---|---|
| | Novel plant genotypes | *Describe the methods by which all novel plant genotypes were produced. This includes those generated by transgenic approaches, gene editing, chemical/radiation-based mutagenesis and hybridization. For transgenic lines, describe the transformation method, the number of independent lines analyzed and the generation upon which experiments were performed. For gene-edited lines, describe the editor used, the endogenous sequence targeted for editing, the targeting guide RNA sequence (if applicable) and how the editor was applied.* |
| | Authentication | *Describe any authentication procedures for each seed stock used or novel genotype generated. Describe any experiments used to assess the effect of a mutation and, where applicable, how potential secondary effects (e.g. second site T-DNA insertions, mosiacism, off-target gene editing) were examined.* |

