## [Peer Review File · Nature Human Behaviour]

Inequality in infrastructure access and its association with health disparities

Corresponding Author: Professor Chuan Liao

Version 0:

Decision Letter:

8th October 2024

Dear Professor Liao,

Thank you once again for your manuscript, entitled "Global inequality in infrastructure access and the associated disparity in health outcomes," and for your patience during the peer review process.

Your manuscript has now been evaluated by 3 reviewers, whose comments are included at the end of this letter. Although the reviewers find your work to be of interest, they also raise some important concerns. We are interested in the possibility of publishing your study in *Nature Human Behaviour*, but would like to consider your response to these concerns in the form of a revised manuscript before we make a decision on publication.

To guide the scope of the revisions, the editors discuss the referee reports in detail within the team, including with the chief editor, with a view to (1) identifying key priorities that should be addressed in revision and (2) overruling referee requests that are deemed beyond the scope of the current study. We hope that you will find the prioritised set of referee points to be useful when revising your study. Please do not hesitate to get in touch if you would like to discuss these issues further.

In particular, we ask you to address the following (as well as all other reviewers' concerns):

- 1) Please improve the literature review, in particular for the relationship between infrastructure access and health. Detail the contribution over prior work.
- 2) To enhance the comprehensive understanding of research questions, we encourage that you analyse the interdependencies between different types of infrastructure access and the association with health.
- 3) Please address reviewer concerns about the clarity and comprehensiveness of your infrastructure measures, including expanding your focus to include water supply and waste management infrastructure (as recommended by Reviewer 2) and ensuring that your approach to infrastructure categorisation is clear and justified (as highlighted by Reviewer 3).
- 4) Please remove all causal language from describing descriptive or correlational findings.
- 5) It is journal policy that p-values larger than 0.05 must not be interpreted as evidence of an effect, unless you a priori specified an alternative threshold. We therefore ask that you revise to remove all interpretations of effects with $p > 0.05$.

In sum, we invite you to revise your manuscript taking into account all reviewer and editor comments. We are committed to providing a fair and constructive peer-review process. Do not hesitate to contact us if there are specific requests from the reviewers that you believe are technically impossible or unlikely to yield a meaningful outcome.

We hope to receive your revised manuscript within two months. I would be grateful if you could contact us as soon as possible if you foresee difficulties with meeting this target resubmission date.

- Include a "Response to the editors and reviewers" document detailing, point-by-point, how you addressed each editor and referee comment. If no action was taken to address a point, you must provide a compelling argument. When formatting this document,

please respond to each reviewer comment individually, including the full text of the reviewer comment verbatim followed by your response to the individual point. This response will be used by the editors to evaluate your revision and sent back to the reviewers along with the revised manuscript.

- Highlight all changes made to your manuscript or provide us with a version that tracks changes.

Link Redacted

We look forward to seeing the revised manuscript and thank you for the opportunity to review your work. Please do not hesitate to contact me if you have any questions or would like to discuss these revisions further.

Sincerely,

Nature Human Behaviour

Reviewer expertise:

Reviewer #1: urban sustainability, environmental justice and health, sustainable infrastructure

Reviewer #2: sustainable infrastructure, climate mitigation

Reviewer #3: urban health, built environment

REVIEWER COMMENTS:

Reviewer #1 (Remarks to the Author):

Thank you for the opportunity to review this manuscript. This global analysis of infrastructure systems and the impacts of infrastructure inequalities on health outcomes by country is impressive and provides novel insight into the importance of access to infrastructure for health and well-being. Overall, I found the study to be sound and the manuscript to clearly document the methods (though these methods are outside of my expertise), report the findings, and contextualize the results within previous research. However, I offer a few suggestions below for further strengthening the study:

- the research significance of the study could be elevated throughout the manuscript, starting with the introduction. The authors emphasize the study's global scale of analysis and the breakdown of different types of infrastructure systems, yet the study's main contribution seems to be in the analysis of how different types of infrastructures and their inequities impact health. This part could be elaborated in greater detail. In the current manuscript, the summary of the contribution/research gap is fairly limited (see lines 60-73). This research contribution could be revisited in the discussion - how do the findings from this extensive study affirm, contradict, or extend previous research?
- the study addresses access to infrastructure in terms of spatial proximity and develops a population-weighted exposure model. While spatial proximity is one common measure of access, there are other critical dimensions of access that should be acknowledged, at least in the limitations section. For instance, living near a facility does not guarantee that a resident living nearby can use the facility. Other research has also pointed out how living near some infrastructure systems has a negative impact on nearby communities (e.g. air pollution from highways).
- the abstract and the first paragraph of the introduction emphasize human decision making around infrastructure, setting up the study to be an examination of how decisions about infrastructure systems impact equity and access. However, this is not the focus of the study. A stronger introduction would better frame the research gap and need for the study.

Reviewer #2 (Remarks to the Author):

The authors of this paper provide a global assessment of infrastructure access and inequalities across social, economic and environmental dimensions, and classify countries and regions according to their access in each of these categories using the established Gini coefficient. They integrate existing critical infrastructure datasets along with additional social and environmental data to map spatial access to infrastructure services at national and sub-national scales, with an added focus on impacts on health indicators. The authors have done a commendable job to undertake this comprehensive analysis, which represents an advance in current knowledge. Such a study is important given the role of equitable access to infrastructure services in achieving global sustainable development outcomes including the SDGs. The use of complementary statistical methods and tests adds robustness to the global infrastructure mapping.

There are however a few areas in the paper that would benefit from further clarification before acceptance for publication:

1. The analysis quantifies both the differences in infrastructure access, as well as inequality in human access to infrastructure, across types and scales. I find it a bit confusing to distinguish these using the current framing, since differences in access within a country or region seems to imply inequality of access. This becomes a bit clearer when reading through the equation derivation in the supplementary material, but the differences between spatial distribution/accessibility and inequality in the context of infrastructure should be discussed upfront in plainer terms.
2. To simplify the spatial analysis of the critical infrastructure dataset the authors have classified the 39 infrastructure types into economic, social, and environmental infrastructure. However, the importance of interdependencies has not been addressed. For example, many infrastructure assets will not be beneficial without accompanying and resilient energy supply networks, transport systems are required to provide access to social facilities, and so on. While I understand it may be out of the study scope, there should be some discussion of the role this may play in affecting access rates.
3. In addition, given the large role that water supply and waste management in economic (energy generation, cooling and industrial waste, among others) and social (e.g. hygiene and health), I wonder if these should also be included in these categories in some way.
4. The impact on health outcomes is the main application of the results, but the study jumps straight into the regression with no background literature, as would be done in most empirical analysis, making it feel a bit disjointed. It would be useful to set this part up with some of the literature on infrastructure access and health, showing how different categories of infrastructure are understood to impact human health and through what mechanisms. This can then be tested empirically through the access metrics generated for the study.

Minor comments:

Line 18: A bit strong to say they directly affected health, perhaps reword in terms of there being a significant correlation or similar.

Line 52/67: Clarify what is meant by a single aspect/dimension.

Line 229: Can you shed light on why these types of countries have well-developed environmental infrastructure? Do they generally have well-developed water and waste systems, or is this measure skewed by the fact that there may be more greenspace, less pollution etc. due to a lack of industrialisation or urbanisation?

Line 412: Clarify how max infrastructure amount by type is calculated. Is it number of assets, capacity, etc.?

Line 268: What is meant by civic opportunities, and what is its significance in the context of the health analysis?

Line 295-299: This should be brought up and discussed in more detail earlier in the results or discussion.

Line 774 (Figure 3): Do the colours imply a prioritisation of one infrastructure type over another? For example, H-H-M over M-H-H. If so, state why in the methods/results – especially since you go on to find that social infrastructure access is correlated with higher life expectancy, something that would also presumably be linked to environmental infrastructure factors such as hygiene and pollution levels.

Reviewer #3 (Remarks to the Author):

Abstract

The term infrastructure should be defined earlier on. Authors can expand by saying “Economic, social and environmental infrastructures stand as fundamental pillars of human and social development.” The word infrastructure alone often refers to physical and this article captures multiple dimensions and should be stated upfront.

Line 60-62: Authors should expand or simplify this sentence. For example, what inequalities are exacerbated? What are examples of ecological complications and/or disruptions to sociocultural norms?

Similarly, “disparities in infrastructure distribution” is stated in line 64. What kind of infrastructure?

Line 74-82. I would recommend bringing this section earlier so readers understand what is meant by infrastructure. Again, infrastructure alone is often thought of as physical infrastructure.

It could be argued that ‘transport’ would fit better within the environmental infrastructure. How were some of these definitions derived? Are there some references the authors can provide here for the definitions they provide.

Lines 110-115. Authors refer to Global North and Global South differences, but then go on to discussed country-level differences that again refer to Global North and South. What is the difference between these two findings?

Line 125-127. When describing the the H-M-L categories in the results section, a reminder of what each of these letters corresponds to is needed. For example "H-M-L"

Line 206. I suggest expanding the sentence to include cities in addition to neighborhoods as some of these investments and policies at the federal or national level impact cities and create disparities among cities.

I didn't see a limitation section. What are some limitations associated with using some of these larger datasets particularly in terms of reporting biases, missing data and data harmonization.

Methods.

I would recommend the authors placing all the datasets into a table to allow readers to reduce text but also provide a better overview.

Some of their references are incomplete and missing information including url information or the name of the report.

Version 1:

Decision Letter:

Our ref: NATHUMBEHAV-24062456A

19th December 2024

Dear Dr. Liao,

Thank you for submitting your revised manuscript "Global inequality in infrastructure access and associated disparity in health outcomes" (NATHUMBEHAV-24062456A). It has now been seen by the original referees and their comments are below. As you can see, the reviewers find that the paper has improved in revision. We will therefore be happy in principle to publish it in Nature Human Behaviour, pending minor revisions to satisfy the referees' final requests and to comply with our editorial and formatting guidelines.

We are now performing detailed checks on your paper and will send you a checklist detailing our editorial and formatting requirements early in the new year. Please do not upload the final materials and make any revisions until you receive this additional information from us.

Sincerely,

Nature Human Behaviour

Reviewer #1 (Remarks to the Author):

Thank you for the opportunity to review the revised manuscript. I appreciate the time and effort the authors put towards the revisions, and they have addressed many of the concerns raised in the first review. I have a few additional suggestions to strengthen the manuscript, mostly based on the writing in the introduction and discussion sections, to highlight the contributions of the study.

The revisions to the introduction provide a stronger rationale for the article, but the flow and organization could be improved. For example:

- The paragraph starting on line 35 (Given the broad interpretation of infrastructure) helps the define key terms, but it might be better placed later in the introduction, following the rationale and premise of the study. For instance, the previous paragraph ends with a statement about "significant knowledge gaps" on health and infrastructure, and some elaboration on these gaps would be helpful as the next paragraph before introducing the study (and how the study addresses these gaps).
- the article could more clearly lay out the knowledge gaps, directly link the literature reviews to the knowledge gaps, and then directly state the two key contributions of the study. Based on my reading, I see these contributes as 1) comparing infrastructural inequalities by country and 2) measuring health impacts of infrastructure inequalities.
- the following sentence on line 90 seems too broad and sweeping, yet is critical to the research contribution of the study: "Furthermore, while access to infrastructure has been extensively studied, few studies have explored how inequality in infrastructure access correlates with health outcomes." There is substantial research on inequities in access to public health

infrastructure and health outcomes, to water infrastructure inequities and health outcomes, etc. This sentence needs to be revised and the associated literature review needs to be refined.

The discussion section is improved, and the addition of the limitations section is helpful. I recognize that this study is an exciting, large-scale, global study of infrastructure across multiple types of infrastructure, and I especially appreciate the focus on equitable access to infrastructure, however, based on the discussion section, the results are not particularly novel and they seem fairly intuitive (equitable access to infrastructure is associated with positive health outcomes, or see line 297 "Key infrastructure components, such as access to transportation, healthcare facilities, and housing, are essential in creating environments that are conducive to good living conditions"). Each of the points raised in the discussion are described as reflecting previous research. Can the authors better highlight the novel contribution of this study? What can this study tell us that previous work did not?

Reviewer #2 (Remarks to the Author):

The authors have done a thorough job of addressing feedback and improving the robustness of various aspects of the study. I am satisfied that all comments have now been addressed.

Reviewer #3 (Remarks to the Author):

The authors have done a thorough job of addressing my concerns. I have no further comments.

Reviewer #3 (Remarks on code availability):

I am unsure what this is in reference to but I did see the authors provided sources for the data they used and descriptions of this data.

Version 2:

Decision Letter:

Dear Professor Liao,

We are pleased to inform you that your Article "Inequality in infrastructure access and its association with health disparities", has now been accepted for publication in Nature Human Behaviour.

When you receive the proofs, we would be grateful if you could please remove asterisks from Table 2 and revise table notes accordingly. To explain, you have reported p-values in the Table. It is also unconventional to use * when $p < .001$, thus it could be misleading for general audience.

Authors may need to take specific actions to achieve [compliance with funder and institutional open access mandates](https://www.springernature.com/gp/open-research/funding/policy-compliance-faqs). If your research is supported by a funder that requires immediate open access (e.g. according to [Plan S principles](https://www.springernature.com/gp/open-research/plan-s-compliance)) then you should select the gold OA route, and we will direct you to the compliant route where possible. For authors selecting the subscription publication route, the journal's standard licensing terms will need to be accepted, including [self-archiving policies](https://www.springernature.com/gp/open-research/policies/journal-policies). Those licensing terms will supersede any other terms that the author or any third party may assert apply to any version of the manuscript.

We welcome the submission of potential cover material (including a short caption of around 40 words) related to your manuscript; suggestions should be sent to Nature Human Behaviour as electronic files (the image should be 300 dpi at 210 x 297 mm in either TIFF or JPEG format). Please note that such pictures should be selected more for their aesthetic appeal than for their scientific

content, and that colour images work better than black and white or grayscale images. Please do not try to design a cover with the Nature Human Behaviour logo etc., and please do not submit composites of images related to your work. I am sure you will understand that we cannot make any promise as to whether any of your suggestions might be selected for the cover of the journal.

With best regards,

Nature Human Behaviour

P.S. Click on the following link if you would like to recommend Nature Human Behaviour to your librarian
<http://www.nature.com/subscriptions/recommend.html#forms>

** Visit the Springer Nature Editorial and Publishing website at http://editorial-jobs.springernature.com?utm_source=ejp_NHumB_email&utm_medium=ejp_NHumB_email&utm_campaign=ejp_NHumB for more information about our career opportunities. If you have any questions please click [here](mailto:editorial.publishing.jobs@springernature.com).

Open Access This Peer Review File is licensed under a Creative Commons Attribution 4.0 International License, which permits use, sharing, adaptation, distribution and reproduction in any medium or format, as long as you give appropriate credit to the original author(s) and the source, provide a link to the Creative Commons license, and indicate if changes were made. In cases where reviewers are anonymous, credit should be given to 'Anonymous Referee' and the source. The images or other third party material in this Peer Review File are included in the article's Creative Commons license, unless indicated otherwise in a credit line to the material. If material is not included in the article's Creative Commons license and your intended use is not permitted by statutory regulation or exceeds the permitted use, you will need to obtain permission directly from the copyright holder.

Response letter

Reviewer #1 (Remarks to the Author):

Thank you for the opportunity to review this manuscript. This global analysis of infrastructure systems and the impacts of infrastructure inequalities on health outcomes by country is impressive and provides novel insight into the importance of access to infrastructure for health and well-being. Overall, I found the study to be sound and the manuscript to clearly document the methods (though these methods are outside of my expertise), report the findings, and contextualize the results within previous research. However, I offer a few suggestions below for further strengthening the study:

> We sincerely appreciate your positive feedback and valuable suggestions on our work. We have carefully revised our manuscript to better present the introduction and discussion. Please find our detailed responses and actions addressing your specific comments below.

1. the research significance of the study could be elevated throughout the manuscript, starting with the introduction. The authors emphasize the study's global scale of analysis and the breakdown of different types of infrastructure systems, yet the study's main contribution seems to be in the analysis of how different types of infrastructures and their inequities impact health. This part could be elaborated in greater detail. In the current manuscript, the summary of the contribution/research gap is fairly limited (see lines 60-73). This research contribution could be revisited in the discussion - how do the findings from this extensive study affirm, contradict, or extend previous research?

> Thank you for the insightful comment and suggestion. We have strengthened this section by consolidating a comprehensive review of the associations between various infrastructure types and human health. Specifically, we highlight three key research gaps in the literature: (1) lack of comparisons across multiple infrastructure types, (2) limited spatial scale of studies (primarily regional), and (3) insufficient focus on examining inequality in infrastructure access relative to access itself. In the revised manuscript, this elaboration has been added to the introduction section to emphasize knowledge gaps and the need for further research (lines 81-97).

“Infrastructure systems can affect human health and well-being through multiple pathways³⁸. For instance, energy supply outages after extreme events are associated with increased mortality risk^{39,40}. Transportation infrastructure also impacts health, with some

being beneficial (e.g., providing physical connectivity) and others detrimental (e.g., traffic noise and injury risks)⁴¹. Social infrastructure, such as healthcare facilities, are positively linked to public health^{42, 43}, and environmental infrastructure like greenspaces are widely seen as beneficial for physical and mental health^{44, 45}. Nonetheless, most of these studies focus on single infrastructure types (whether economic, social, or environmental), and research on health disparities across various infrastructure types is often limited to a regional scale^{46, 47}. Furthermore, while access to infrastructure has been extensively studied, few studies have explored how inequality in infrastructure access correlates with health outcomes. Understanding how disparities in infrastructure access translate into health inequality is crucial, as infrastructure inequities often reflect and reinforce broader health disparities^{36, 46, 48}. Addressing this issue requires research focusing on measuring inequality across multiple infrastructure types to reveal how access disparities correlate with health outcomes across populations, which will shed light on targeted strategies to improve infrastructure equity, optimize resource distribution, and promote sustainable development and human well-being.”

In the discussion section of the revised manuscript, we further summarize this study’s contributions as follows: (1) contrasting disparities in infrastructure access and inequality between the Global North and South; (2) joint assessment of multi-dimensional and cross-regional infrastructure access differences; (3) quantification of human-infrastructure interactions; and (4) exploration of associations between infrastructure and human health. Particularly, in response to your suggestion regarding the fourth contribution, we discuss how our findings complement and extend previous studies, as well as the potential implications for policymaking (lines 252-266 and 296-304).

“Our analysis also sheds light on the linkages between infrastructure and human health, underscoring that equitable access to infrastructure, particularly in economic and social dimensions, is fundamental for enhancing human health and well-being. At the country scale, we found a positive relationship between infrastructure access and HALE and a negative relationship between infrastructure access inequality and HALE (Fig. 5). In the United States, similar patterns appeared at the county scale, where greater infrastructure access values corresponded to higher life expectancy and lower disease mortality rates (Supplementary Fig. 6). These findings are consistent with previous studies on the relationship between individual types of infrastructure and health outcomes across regions^{39, 40, 41, 42, 43, 44, 45}. Our mixed-effects regression modelling further allowed for a quantitative comparison of health disparities across different infrastructure types and between levels of access and inequality. Notably, the coefficient for economic

infrastructure inequality (GiniEco) showed the largest absolute value compared to economic infrastructure access (ExpEco) and social infrastructure access (ExpSoc) (Table 2), highlighting the key role of infrastructure inequality—a factor largely underexplored in the literature.”

“Lastly, our analysis emphasizes the importance of reducing infrastructure disparities to improve health outcomes, with a specific focus on economic and social infrastructure. Key infrastructure components, such as access to transportation, healthcare facilities, and housing, are essential in creating environments that are conducive to good living conditions. Our results demonstrate that disparities in economic and social infrastructure are closely associated with life expectancy. This indicate that even small increases in inequality can compromise health outcomes. By implementing policies that secure universal and inclusive infrastructure access, policymakers can foster healthier and more resilient communities, further contributing to the global goal of improved well-being and quality of life.”

2. the study addresses access to infrastructure in terms of spatial proximity and develops a population-weighted exposure model. While spatial proximity is one common measure of access, there are other critical dimensions of access that should be acknowledged, at least in the limitations section. For instance, living near a facility does not guarantee that a resident living nearby can use the facility. Other research has also pointed out how living near some infrastructure systems has a negative impact on nearby communities (e.g. air pollution from highways).

> Thank you for the good point. We agree that spatial proximity does not necessarily translate into access. We have acknowledged this issue in the discussion section (lines 313-324).

“Second, the population-weighted exposure model used in this study assumes static population distributions across space and time, focusing on physical access to nearby infrastructure rather than accounting for spatiotemporal interactions between infrastructure and mobile individuals. However, as people move throughout their daily lives, they may benefit from distant infrastructure (e.g., hospitals and schools) or experience negative impacts from nearby infrastructure (e.g., noise and air pollution from highways). Moreover, spatial proximity does not always guarantee higher accessibility due to spatial segregation, such as gated communities or restricted access roads and facilities^{61, 62, 63, 64}. To capture these dynamics more realistically, future studies should consider diverse types of infrastructure accessibility beyond mere spatial proximity and

incorporate human mobility data to assess human-centric exposure through a spatiotemporally explicit interaction framework.”

3. the abstract and the first paragraph of the introduction emphasize human decision making around infrastructure, setting up the study to be an examination of how decisions about infrastructure systems impact equity and access. However, this is not the focus of the study. A stronger introduction would better frame the research gap and need for the study.

> We apologize if the previous framing in the abstract and introduction unintentionally suggested a focus on human decision-making processes rather than on infrastructure access and equity impacts. To avoid confusion, we have revised the abstract and introduction to emphasize the primary research gap and contribution—the need for a global examination of multi-infrastructure access, inequality, and related health disparities. The structure of the introduction has been reorganized as follows: first, we highlight the importance of infrastructure, the overall research gap, the need for this study, and our overarching goal. We then define key terms, including infrastructure, infrastructure access, and infrastructure access inequality. Next, we review prior studies in detail on infrastructure access measurement, infrastructure access inequality assessment, and the association between infrastructure and human health, summarizing key research gaps in each area. Finally, we restate the aim and contribution of this paper by outlining three research questions. We hope these revisions better situate the study within the existing literature and clarify its intended contribution.

Reviewer #2 (Remarks to the Author):

The authors of this paper provide a global assessment of infrastructure access and inequalities across social, economic and environmental dimensions, and classify countries and regions according to their access in each of these categories using the established Gini coefficient. They integrate existing critical infrastructure datasets along with additional social and environmental data to map spatial access to infrastructure services at national and sub-national scales, with an added focus on impacts on health indicators. The authors have done a commendable job to undertake this comprehensive analysis, which represents an advance in current knowledge. Such a study is important given the role of equitable access to infrastructure services in achieving global sustainable development outcomes including the SDGs. The use of complementary statistical methods and tests adds robustness to the global infrastructure mapping.

> Thank you very much for the detailed assessment and for recognizing our work positively. Based on your constructive suggestions, we have carefully revised our manuscript to better present the introduction, results, discussion, and methods.

There are however a few areas in the paper that would benefit from further clarification before acceptance for publication:

> Please find our point-by-point responses to your specific comments below.

1. The analysis quantifies both the differences in infrastructure access, as well as inequality in human access to infrastructure, across types and scales. I find it a bit confusing to distinguish these using the current framing, since differences in access within a country or region seems to imply inequality of access. This becomes a bit clearer when reading through the equation derivation in the supplementary material, but the differences between spatial distribution/accessibility and inequality in the context of infrastructure should be discussed upfront in plainer terms.

> Thank you for bringing this issue of key term definition to our attention. Infrastructure access refers to people's proximity to nearby infrastructure, while infrastructure access inequality highlights disparities in this access within regions (e.g., countries). Now, at the beginning of the introduction for the revised manuscript, we have included definitions for infrastructure, infrastructure access, and infrastructure access inequality (lines 35-40). We believe these clarifications enhance the overall clarity of our paper.

“Given the broad interpretation of infrastructure (see discussions on its definition and classification in the Supplementary Information), here, we define infrastructure as the physical systems and facilities essential for the functioning of the economy and society, infrastructure access as individuals’ capacity to reach and utilize nearby infrastructure, and infrastructure access inequality as geographical disparities in the availability of these systems and facilities.”

2. To simplify the spatial analysis of the critical infrastructure dataset the authors have classified the 39 infrastructure types into economic, social, and environmental infrastructure. However, the importance of interdependencies has not been addressed. For example, many infrastructure assets will not be beneficial without accompanying and resilient energy supply networks, transport systems are required to provide access to social facilities, and so on. While I understand it may be out of the study scope, there should be some discussion of the role this may play in affecting access rates.

> Thank you for this insightful comment. We agree that interdependences between infrastructure types can play an important role in shaping access rates, realistic benefit, and associated health outcomes. To explore this, we first calculated Pearson correlation coefficients for human access to each of the seven critical infrastructure (CI) systems at the country level (Figure R1). The average correlation across these systems was 0.45, with transportation-health, energy-transportation, and transportation-education access showing high correlations at 0.67, 0.64, and 0.55, respectively. In contrast, water and waste infrastructures exhibited lower correlations with other CI systems.

Figure R1. Heatmap of the Pearson correlation coefficients between access to CI systems.

Next, we examined the association between infrastructure interdependencies and healthy life expectancy (HALE), focusing on energy, transportation, and health infrastructure given their relatively high correlations. In addition to examining individual access variables, we included interaction terms for each pair of infrastructure access variables in our regression models. For example, the term *Energy*Transportation* represents combined access to energy and transportation infrastructure. The coefficients of these interaction terms indicate whether the combined impact of the two infrastructures on health outcomes is stronger (>0) or weaker (<0) than the sum of their individual effects.

Our results showed that greater access to energy, transportation, and health infrastructure alone was linked to improved health outcomes (Table R1). Regarding their combined effects, we found that the interaction between energy and transportation infrastructure access was negatively associated with health, suggesting a weaker influence compared to their individual effects. Conversely, the joint impact of transportation and health

infrastructure was significantly associated with increased HALE, indicating that people with greater access to both tend to have longer life expectancy. However, no significant correlation was found between the interaction of energy and health infrastructure access and HALE.

Table R1. Association between energy, transportation, and health infrastructure access and human health. Linear mixed effects regression models are constructed by taking healthy life expectancy (HALE) as the response variable. * denotes a significance level of $p < 0.05$.

Response variable	HALE			
Energy	8.55* (2.33)	6.0* (1.76)		13.22* (2.85)
Transportation	13.66* (2.53)		8.56* (1.98)	10.80* (3.19)
Health		8.28* (2.37)	5.42* (2.47)	5.82* (3.03)
Energy*Transportation	-12.05* (3.61)			-15.21* (4.95)
Energy*Health		-6.09 (3.57)		-8.15 (5.11)
Transportation*Health			10.88* (3.60)	14.92* (5.13)
LnPop	-1.78* (0.27)	-2.14* (0.29)	-2.03* (0.27)	-1.78* (0.27)
LnGDP	1.51* (0.30)	2.17* (0.31)	1.96* (0.29)	1.47* (0.30)
const	61.53* (2.90)	61.86* (3.11)	62.41* (3.09)	62.81* (2.93)
R ²	0.66	0.61	0.62	0.68

While this analysis goes beyond the scope of our current study, it suggests an area for further investigation, that is, understanding the collective impact of infrastructure interdependencies on human health. We have highlighted this point in the discussion section of the revised manuscript (lines 324-330).

“Third, although this research reveals health disparities associated with various infrastructure types, it does not consider the effect of interdependencies among them⁶⁵. For example, many critical infrastructure sectors cannot operate if energy infrastructure is not functioning⁶⁶. This is further supported by our correlation analysis, where access to transportation, energy, and health infrastructure exhibits higher correlations compared to other types (Supplementary Fig. 7). In the next stage, more work is needed to understand how these interdependencies collectively impact human health.”

Accordingly, we have also complemented this exploratory analysis of interdependencies between infrastructure access and its association with human health in the Supplementary Information.

3. In addition, given the large role that water supply and waste management in economic (energy generation, cooling and industrial waste, among others) and social (e.g. hygiene and health), I wonder if these should also be included in these categories in some way.

> Thank you for your suggestion. Our economic-social-environmental infrastructure framework is designed to be flexible, which can accommodate a wide range of critical infrastructure systems based on different research and application needs. To illustrate this adaptability, we have incorporated water and waste infrastructure into our economic and social infrastructure categories and compared the country-level access values with previous versions (Figure R2). In these plots, each point represents a single country, with the x-axis showing the combined economic/social, water, and waste infrastructure access values, and the y-axis representing economic/social infrastructure access alone. The results reveal a high correlation between them, with R^2 values of 0.97 for economic infrastructure and 0.95 for social infrastructure. This indicates that the inclusion of water and waste infrastructure has a limited impact on the overall socioeconomic access metrics, probably due to the relatively low density of these systems compared to other critical infrastructure systems (Figure R3). Statistically, most pixels (> 95%) in the water and waste infrastructure data have a value of 0 (Table R2).

Figure R2. Scatter plots comparing the access values of economic/social, water, and waste infrastructure with economic/social infrastructure alone at the country level.

Figure R3. Maps of critical infrastructure distribution for energy, transportation, telecommunication, health, education, water, and waste systems (a-g). For each system, the number of infrastructure types was normalized to a range of 0-1 and then averaged (see Supplementary Table 5 for the correspondence between infrastructure systems and types). (h) shows the normalized average across all 39 infrastructure types.

Table R2. Classification and quantile distribution of the global critical infrastructure data.

Category	CI system	Infrastructure type	Quantiles						
			50%	60%	70%	80%	90%	95%	100%
Economic	Energy	Cable	0	0	0	0	0	0	269
Economic	Energy	Line	0	0	0	0	10	18	260
Economic	Energy	Minor Line	0	0	0	0	0	0	204
Economic	Energy	Plant	0	0	0	0	0	0	42
Economic	Energy	Power Pole	0	0	0	0	0	0	17828
Economic	Energy	Power Tower	0	0	0	0	24	53	1157
Economic	Energy	Substation	0	0	0	0	0	0	1
Economic	Transportation	Airports	0	0	0	0	0	0	79
Economic	Transportation	Primary	0	0	0	0	11	18	529
Economic	Transportation	Railway	0	0	0	0	0	9	805
Economic	Transportation	Secondary	0	0	0	0	9	15	361
Economic	Transportation	Tertiary	0	4	14	31	74	130	2901
Economic	Telecommunication	Communication Tower	0	0	0	0	0	0	401
Economic	Telecommunication	Mast	0	0	0	0	0	0	131
Social	Health	Alternative	0	0	0	0	0	0	72
Social	Health	Birthing Center	0	0	0	0	0	0	6
Social	Health	Blood Donation	0	0	0	0	0	0	5
Social	Health	Clinic	0	0	0	0	0	0	481
Social	Health	Dentist	0	0	0	0	0	0	455
Social	Health	Doctors	0	0	0	0	0	0	865
Social	Health	Hospital	0	0	0	0	0	0	585
Social	Health	Laboratory	0	0	0	0	0	0	101
Social	Health	Optometrist	0	0	0	0	0	0	15
Social	Health	Pharmacy	0	0	0	0	0	0	996
Social	Health	Physiotherapist	0	0	0	0	0	0	105
Social	Health	Rehabilitation	0	0	0	0	0	0	13
Social	Education	College	0	0	0	0	0	0	14
Social	Education	Kindergarten	0	0	0	0	0	0	1
Social	Education	Library	0	0	0	0	0	0	1
Social	Education	School	0	0	0	0	0	0	10
Social	Education	University	0	0	0	0	0	0	23
Environmental	Water	Reservoir	0	0	0	0	0	0	105
Environmental	Water	Reservoir Covered	0	0	0	0	0	0	9
Environmental	Water	Water Tower	0	0	0	0	0	0	1
Environmental	Water	Water Well	0	0	0	0	0	0	1
Environmental	Water	Water Works	0	0	0	0	0	0	4
Environmental	Waste	Landfill	0	0	0	0	0	0	90
Environmental	Waste	Wastewater Treatment Plant	0	0	0	0	0	0	13
Environmental	Waste	Waste Transfer Station	0	0	0	0	0	0	7

Since the inclusion of water and waste infrastructure does not alter our major findings and conclusions, we have retained the original definitions of economic and social infrastructure, along with their associated access and inequality measures, in the revised manuscript. We have also highlighted this adaptability in the discussion section (lines 305-313) and added the sensitivity analysis in the Supplementary Information.

“This research also acknowledges several levels of limitations that suggest directions for future investigations. First, while we examine infrastructure from economic, social, and environmental perspectives, it is important to note that infrastructure classifications are diverse, with no consistent standards in the literature^{2,58}. In some cases, categories may overlap; for instance, water and waste infrastructure can be considered as economic infrastructure^{59,60}. Our focus, however, is on developing a flexible comparative framework, which can accommodate a wide range of critical infrastructure systems and effectively reveal access inequalities across different types. Future research could adapt this model to fit various infrastructure classifications based on specific application needs.”

4. The impact on health outcomes is the main application of the results, but the study jumps straight into the regression with no background literature, as would be done in most empirical analysis, making it feel a bit disjointed. It would be useful to set this part up with some of the literature on infrastructure access and health, showing how different categories of infrastructure are understood to impact human health and through what mechanisms. This can then be tested empirically through the access metrics generated for the study.

> Thanks for pointing this out. We have added a comprehensive literature review in the introduction section, where we discussed the associations between various infrastructure types and human health, and summarized three key research gaps (lines 81-97). Additionally, we have compared our findings to previous studies and highlighted this research’s contributions and implications in the discussion section (lines 252-266 and 296-304). For more details, please refer to our response to Reviewer #1’s Comment 1.

Minor comments:

5. Line 18: A bit strong to say they directly affected health, perhaps reword in terms of there being a significant correlation or similar.

> Thank you for this suggestion. The sentence now reads: *“We find that both infrastructure access and inequality are linked to human health outcomes, with this association being*

especially pronounced in economic infrastructure inequality” (lines 16-18). We have also reviewed the entire manuscript thoroughly to avoid any causal language such as “affect”, “contribute to”, or “lead to”.

6. Line 52/67: Clarify what is meant by a single aspect/dimension.

> Many thanks. We have revised these sentences as “*Additionally, existing evaluation methods largely focus on a single infrastructure system (e.g., coastal built assets)*” (lines 57-58) and “*However, previous studies of infrastructure inequality have predominantly focused on evaluating a single dimension (e.g., inequality in water infrastructure)*” (lines 75-76).

7. Line 229: Can you shed light on why these types of countries have well-developed environmental infrastructure? Do they generally have well-developed water and waste systems, or is this measure skewed by the fact that there may be more greenspace, less pollution etc. due to a lack of industrialisation or urbanisation?

> Thanks for raising this good point. We calculated the access levels for each of the five components (water, waste, greenspace, air pollution, and heat) that make up our environmental infrastructure. We then ranked countries based on these individual values and compared the rankings with the overall environmental infrastructure score. The results indicate that heat, air pollution, and greenspace demonstrate high correlations with environmental infrastructure access, with R^2 values of 0.56, 0.49, and 0.29, respectively (Figure R4c-e). This suggests that countries with lower heat and air pollution exposure and better greenspace access are more likely to have well-developed environmental infrastructure. However, there is no strong correlation between access to water and waste infrastructure and overall environmental infrastructure (Figure R4a-b). As noted earlier, this is mainly due to limited coverage and lower values for water and waste infrastructure in the global critical infrastructure data (Figure R3 and Table R2). The sentence has now been revised as: “*These countries have relatively well-developed environmental infrastructure, primarily due to lower exposure to heat and air pollution and greater access to greenspace (Supplementary Fig. 4); however, they are poorly served in the economic and social dimensions, making the promotion of socioeconomic development a primary focus for the future*” (lines 229-233).

Figure R4. Scatter plots comparing the rank of overall environmental infrastructure access with the rank of single environmental factor exposures at the country level: (a) water, (b) waste, (c) greenspace, (d) air pollution, and (e) heat.

8. Line 412: Clarify how max infrastructure amount by type is calculated. Is it number of assets, capacity, etc.?

> Thank you for the question. The maximum infrastructure amount refers to the highest raster value in the critical infrastructure dataset, corresponding to the 100th percentile values in Table R2. This value represents the total quantity of each infrastructure type (e.g., number of power poles) within a given grid cell, rather than capacity. We've clarified this in the revised manuscript (lines 365-370).

“In the original CI dataset, the raster value represents the total amount of infrastructure (e.g., number of power poles) within a given grid cell⁶⁷. Statistics show that some infrastructure types, such as power poles and tertiary roads, have significantly higher quantities than others (Supplementary Table 5). To enable cross-type comparisons, we normalized each infrastructure type layer within each CI system to a 0–1 scale by dividing by the maximum grid value for that type (Supplementary Fig. 8).”

9. Line 268: What is meant by civic opportunities, and what is its significance in the context of the health analysis?

> We apologize for any misunderstanding. We have removed discussions on civic opportunities, and revised the sentence as “*Our analysis also sheds light on the linkages between infrastructure and human health, underscoring that equitable access to infrastructure, particularly in economic and social dimensions, is fundamental for enhancing human health and well-being*” (lines 252-254).

10. Line 295-299: This should be brought up and discussed in more detail earlier in the results or discussion.

> Thank you for your suggestion. Same as response to your Comment 4, we have reviewed studies on the associations between infrastructure and human health in the introduction and summarized our findings and contributions to the literature in the discussion. Therefore, emphasizing the importance of equitable access to various infrastructure systems in promoting human health is now better aligned with the context.

11. Line 774 (Figure 3): Do the colours imply a prioritisation of one infrastructure type over another? For example, H-H-M over M-H-H. If so, state why in the methods/results – especially since you go on to find that social infrastructure access is correlated with higher life expectancy, something that would also presumably be linked to environmental infrastructure factors such as hygiene and pollution levels.

> We use different color schemes to render these categories, representing varying infrastructure access levels and disparities across the economic, social, and environmental dimensions. However, our intent is not to prioritize one infrastructure type over another but rather to depict various combinations of high (H), medium (M), and low (L) access levels. The strength of our framework lies in its flexibility to comparatively highlight disparities across infrastructure types. Noted that human health is only one dimension we show outcomes associated with these disparities, while the framework can also be adapted for practical applications across different contexts. For instance, we observe an 'L-L-H' pattern in countries such as Fiji, Madagascar, Swaziland, Timor-Leste, and Vanuatu, where environmental infrastructure is relatively well-established, yet economic and social dimensions are under-developed, making socioeconomic development a key focus for the future. This information can be extremely helpful for policymakers in identifying regions where access to different infrastructure types is most fragile and

inequitable, helping them target areas in urgent need of support. We have clarified this in the discussion section (lines 218-235).

“The joint assessment of economic-social-environmental infrastructure offers an enriched understanding of global infrastructure access and inequality. By categorizing infrastructure access into high, medium, and low for each dimension, we identify 27 unique categories of combination, which group into three general classes representing varying levels and disparities in economic, social, and environmental infrastructure access. The advantages of our framework lie in its ability to elucidate these levels and disparities across different dimensions, thereby assisting policymakers in more effectively addressing diverse infrastructure challenges^{50, 51}. For instance, patterns like ‘H-H-L’, ‘H-M-L’, or ‘M-H-L’ demonstrate well-developed socio-economic infrastructure but often neglected environmental issues in some countries and regions. Conversely, we observe an ‘L-L-H’ pattern in countries in Africa, Southeast Asia, and the South Pacific Ocean, including Fiji, Madagascar, Swaziland, Timor-Leste, and Vanuatu. These countries have relatively well-developed environmental infrastructure, primarily due to lower exposure to heat and air pollution and greater access to greenspace (Supplementary Fig. 4); however, they are poorly served in the economic and social dimensions, making the promotion of socioeconomic development a primary focus for the future. These patterns provide a comprehensive view of global infrastructure access, pinpointing areas where human access to specific types of infrastructure is most fragile, inequitable, and urgently in need of aid.”

Reviewer #3 (Remarks to the Author):

Abstract

1. The term infrastructure should be defined earlier on. Authors can expand by saying “Economic, social and environmental infrastructures stand as fundamental pillars of human and social development.” The word infrastructure alone often refers to physical and this article captures multiple dimensions and should be stated upfront.

> Thank you for the valuable suggestion. Indeed, this research focuses on the physical components of infrastructure rather than soft infrastructure (such as human capital), as physical infrastructure is more measurable. Our unique contribution lies in adopting a multi-dimensional economic-social-environmental perspective to assess disparities across various infrastructure types and regions. This definition and classification of infrastructure broadly align with previous literature, and we have added a new section (“Definitions and classifications of infrastructure”) in the Supplementary Information to elaborate further. In response, we have revised the beginning of the abstract as: *“Economic, social, and environmental infrastructure forms a fundamental pillar of societal development”* (lines 4-5). Additionally, we have clarified definitions and classifications of infrastructure at the start of the introduction section (lines 35-48).

“Given the broad interpretation of infrastructure (see discussions on its definition and classification in the Supplementary Information), here, we define infrastructure as the physical systems and facilities essential for the functioning of the economy and society, infrastructure access as individuals’ capacity to reach and utilize nearby infrastructure, and infrastructure access inequality as geographical disparities in the availability of these systems and facilities. We categorize infrastructure into three types based on their primary functions: economic, social, and environmental (Fig. 1a). Economic infrastructure encompasses facilities that support business activities, such as telecommunication, energy supply systems, and transport and distribution networks. Social infrastructure comprises facilities providing social services, such as schools and hospitals. Environmental infrastructure, linked to living conditions and ecosystem services, includes resources like water and waste facilities, greenspaces, clean air, and thermal comfort. This multi-dimensional conceptualization of infrastructure leads to an enhanced understanding of the diverse ways in which individuals meet their needs and communities manage public goods across regions.”

2. Line 60-62: Authors should expand or simplify this sentence. For example, what inequalities are exacerbated? What are examples of ecological complications and/or disruptions to sociocultural norms?

> Thanks for this good suggestion. We have expanded the sentence as “*Concerns have recently intensified regarding the adverse effects of infrastructure development, including ecological complications²¹, heightened inequalities²², and disruptions to sociocultural norms²³. For example, large-scale infrastructure projects, such as road construction, can lead to substantial forest loss²⁴. In urban areas, uneven spatial distribution of infrastructure and access to amenities can exacerbate inequalities among various social groups²⁵*” (lines 66-71).

3. Similarly, “disparities in infrastructure distribution” is stated in line 64. What kind of infrastructure?

> The sentence has been revised as “*A number of recent research conducted in India^{26, 27}, Indonesia^{28, 29}, South Africa^{30, 31}, and elsewhere^{32, 33, 34, 35} has demonstrated disparities in infrastructure distribution and accessibility—such as electricity and telecommunications—across cities and communities, thereby highlighting potential threats to urban sustainability and public health^{3, 6, 36, 37}*” (lines 71-74).

4. Line 74-82. I would recommend bringing this section earlier so readers understand what is meant by infrastructure. Again, infrastructure alone is often thought of as physical infrastructure.

> Thank you for the suggestion. In line with our response to your Comment 1, we have moved this section to the beginning of the article and included definitions for infrastructure, infrastructure access, and infrastructure access inequality.

5. It could be argued that ‘transport’ would fit better within the environmental infrastructure. How were some of these definitions derived? Are there some references the authors can provide here for the definitions they provide.

> Thank you for the insightful comment. Although infrastructure is defined and classified in various ways in the literature, it is generally understood as both the fundamental physical and organizational systems essential for the functioning of society and the economy, which can be categorized based on two main criteria: features and functions. The first

criterion emphasizes the structural or material nature of infrastructure (e.g., network or point-based), while the second classifies infrastructure by its role, such as economic/social, core/non-core, or basic/complementary. Transportation, for instance, is generally regarded as a core component of economic infrastructure. Since the economic/social classification is the most widely used, we adopted it in our study. Additionally, recent research underscores the importance of environmental infrastructure in promoting human health, social well-being, and sustainable development. Consequently, we examine infrastructure from economic, social, and environmental perspectives. To support this, we have added a section discussing the “Definitions and classifications of infrastructure” in literature in the Supplementary Information, which is duplicated as follows.

“Definition and classification of infrastructure

*Infrastructure is essential for supporting economic activities, social well-being, and human development. According to the Cambridge Dictionary, infrastructure refers to “the basic systems and services, such as transport and power supplies, that a country or organization uses in order to work effectively”. As early as the 18th century, economists began documenting the socioeconomic nature of infrastructure. Adam Smith, for example, justified the principle of the “invisible hand of the market” while assigning the state the role of infrastructure investor, emphasizing the state’s obligation to maintain public facilities and institutions¹. The term “infrastructure” was first used by the military to describe war logistics during the Second World War and has since expanded to various fields². In his famous book of *The Strategy of Economic Development*, German economist Albert Hirschman defined infrastructure as “capital that provides public services”, highlighting two key elements: capitalness and publicness³. Bühr⁴ further described infrastructure as “the sum of all relevant economic data such as rules, stocks, and measures with the function of mobilizing the economic potentialities of economic agents”. More recently, Fulmer⁵ defined infrastructure as “the physical components of interrelated systems providing commodities and services essential to enable, sustain, or enhance societal living conditions”. Despite its varying definitions, infrastructure is widely recognized as both the fundamental physical and organizational systems necessary for the functioning of society and the economy^{6, 7, 8}. In this article, we focus on the physical components of infrastructure, as they are more tangible and straightforward to measure.*

As reviewed in previous studies, infrastructure can encompass a wide range of types^{6, 7, 8}. Generally, two main criteria are used for classification: features and functions. Under the prior criterion, research often focuses on the structure or material nature of infrastructure. For instance, Biehl⁹ classified infrastructure into network (e.g., roads, railways, electrical

facilities) and point (e.g., schools, hospitals, museums), depending on whether active human involvement is required for the operation of a structure. Similarly, infrastructure was classified into tangible (hard) and intangible (soft) assets based on their physical nature¹⁰.

Another approach classifies infrastructure according to its functional role. Hansen¹¹ distinguished between economic and social infrastructure based on its direct or indirect influence on regional economic development, a classification widely used in the literature^{12, 13, 14, 15}. Researchers have also differentiated between “core” and “non-core” infrastructure based on their importance to the economy’s sustainable functioning. Sturm and Jacobs¹⁶ used a similar distinction between “basic” and “complementary” infrastructure. Jochimsen¹⁷ categorized infrastructure into three kinds: personal (which shapes the values of economic agents), institutional (which promotes social integration), and material (which addresses physical and social needs).

Moreover, some studies have adopted a dual classification approach, considering both features and functions. Vaughan-Morris¹⁸ described infrastructure as “hard” and “soft” types, where hard infrastructure includes economic, social, and industrial facilities, while soft infrastructure refers primarily to institutions and intangible assets, such as government buildings, laws, regulations. Despite variations in classification systems, the majority incorporate critical infrastructure systems that are essential to societal operation, such as transportation, communications, energy supply, healthcare, and sanitation systems^{6, 7, 8}.

In recent years, there has been a growing recognition of the role that the natural environment, such as green and water infrastructure^{19, 20, 21, 22}, plays in human and social development. Frischmann²³ argued that the natural environment functions similarly to traditional infrastructure, serving as a vital input for a wide range of human and natural goods and services. He introduced the concept of “environmental infrastructure”, highlighting its essential role in supporting various processes and services. Consequently, recent studies and reports have called for a broader integration of environmental considerations into infrastructure design and management to maintain human health and social well-being. For instance, the World Bank’s report “Putting Nature to Work: Integrating Green and Gray Infrastructure for Water Security and Climate Resilience” discussed the role of nature-based solutions in boosting urban resilience, reducing pollution, and mitigating climate change²⁴. Likewise, the Organisation for Economic Co-operation and Development (OECD), in its report “Financing Climate Futures: Rethinking Infrastructure”, stressed the importance of investing in green infrastructure and integrating climate considerations into infrastructure planning for long-term sustainability²⁵.

Furthermore, as no standardized infrastructure classification exists, we point out that future research can adopt classifications tailored to specific application needs. We have highlighted this point in the discussion section (lines 305-313).

“This research also acknowledges several levels of limitations that suggest directions for future investigations. First, while we examine infrastructure from economic, social, and environmental perspectives, it is important to note that infrastructure classifications are diverse, with no consistent standards in the literature^{2,58}. In some cases, categories may overlap; for instance, water and waste infrastructure can be considered as economic infrastructure^{59,60}. Our focus, however, is on developing a flexible comparative framework, which can accommodate a wide range of critical infrastructure systems and effectively reveal access inequalities across different types. Future research could adapt this model to fit various infrastructure classifications based on specific application needs.”

6. Lines 110-115. Authors refer to Global North and Global South differences, but then go on to discussed country-level differences that again refer to Global North and South. What is the difference between these two findings?

> Thank you for your question. The second division refers to **county**-level differences in infrastructure access, providing a more granular view of within-country disparities (Figure R5). Here the aim is to highlight the similar patterns observed between country- and county-level results, which validates the robustness of our assessment model.

Figure R5. County-level human access to (a) economic, (b) social, and (c) environmental infrastructure distributions across the globe. Black boundaries indicate countries in the Global South.

7. Line 125-127. When describing the H-M-L categories in the results section, a reminder of what each of these letters corresponds to is needed. For example “H-M-L”

> Thank you for the suggestion. We have revised the text for better clarity (lines 133-135).

“For example, 'H-M-L' indicates 'high-medium-low' levels for economic, social, and environmental infrastructure access, respectively.”

8. Line 206. I suggest expanding the sentence to include cities in addition to neighborhoods as some of these investments and policies at the federal or national level impact cities and create disparities among cities.

> Thanks for this good suggestion. We have revised the sentence as *“While infrastructure investment is essential, economic growth alone cannot resolve infrastructure disparities; in some cases, it may even exacerbate regional disparities due to policies or investments that favour specific neighbourhoods over others^{53, 54}. At the national level, such targeted investments and policies can further widen disparities across regions and cities^{55, 56, 57}, and if population growth outpaces infrastructure construction, disparities in infrastructure access may worsen²⁶”* (lines 281-287).

In the subsequent text, we further discussed their implications for policymaking (lines 287-295): *“To effectively address these challenges, policymakers need to prioritize equitable infrastructure allocation based on a spatial assessment of infrastructure needs across multiple dimensions, regions, and human-infrastructure interaction settings. This can involve implementing redistributive policies that channel resources to historically underserved regions and enforcing guidelines to prevent infrastructure development projects from favouring particular neighbourhoods at the expense of others. Policymakers should also account for real-time population dynamics to ensure that infrastructure expansions can accommodate rising demand, especially in regions where rapid population increases may otherwise outstrip infrastructure improvements”*.

9. I didn't see a limitation section. What are some limitations associated with using some of these larger datasets particularly in terms of reporting biases, missing data and data harmonization.

> Thank you for pointing this out. We have added a limitations section discussing the methodological and data constraints of this study (lines 305-341).

“This research also acknowledges several levels of limitations that suggest directions for future investigations. First, while we examine infrastructure from economic, social, and environmental perspectives, it is important to note that infrastructure classifications are diverse, with no consistent standards in the literature^{2, 58}. In some cases, categories may overlap; for instance, water and waste infrastructure can be considered as economic infrastructure^{59, 60}. Our focus, however, is on developing a flexible comparative framework, which can accommodate a wide range of critical infrastructure systems and effectively reveal access inequalities across different types. Future research could adapt this model to fit various infrastructure classifications based on specific application needs. Second, the population-weighted exposure model used in this study assumes static population distributions across space and time, focusing on physical access to nearby infrastructure rather than accounting for spatiotemporal interactions between infrastructure and mobile individuals. However, as people move throughout their daily lives, they may benefit from distant infrastructure (e.g., hospitals and schools) or experience negative impacts from nearby infrastructure (e.g., noise and air pollution from highways). Moreover, spatial proximity does not always guarantee higher accessibility due to spatial segregation, such as gated communities or restricted access roads and facilities^{61, 62, 63, 64}. To capture these dynamics more realistically, future studies should consider diverse types of infrastructure accessibility beyond mere spatial proximity and incorporate human mobility data to assess human-centric exposure through a spatiotemporally explicit interaction framework. Third, although this research reveals health disparities associated with various infrastructure types, it does not consider the effect of interdependencies among them⁶⁵. For example, many critical infrastructure sectors cannot operate if energy infrastructure is not functioning⁶⁶. This is further supported by our correlation analysis, where access to transportation, energy, and health infrastructure exhibits higher correlations compared to other types (Supplementary Fig. 7). In the next stage, more work is needed to understand how these interdependencies collectively impact human health. Finally, there are data gaps and limitations associated with the infrastructure dataset used in this study. On the one hand, as the critical infrastructure dataset is derived from a voluntary data source of OpenStreetMap⁶⁷, it may contain missing data, particularly in less-developed areas, which limits the representation of certain countries or regions. On the other hand, our inequality analysis is conducted at the national scale, constrained by the dataset's relatively coarse spatial resolution (0.1° × 0.1°). This approach does not capture intra-country heterogeneity in inequality levels, as significant disparities in infrastructure provision can exist even within developed countries with low overall inequality. Moving forward, we plan to integrate multi-source data, such as high-resolution satellite imagery and human mobility data, to

analyse infrastructure exposure inequalities at finer spatial scales and identify vulnerable hotspots requiring targeted policy interventions and initiatives.”

Methods.

10. I would recommend the authors placing all the datasets into a table to allow readers to reduce text but also provide a better overview.

> Thank you for the suggestion. We have consolidated all datasets into a single table (Supplementary Table 4) and moved the detailed data descriptions to the Supplementary Information section for improved clarity in the main text.

11. Some of their references are incomplete and missing information including url information or the name of the report.

> Thanks for spotting this issue. We have reviewed and updated all the references to ensure they align with the journal’s required format.

Response letter

Reviewer #1 (Remarks to the Author):

Thank you for the opportunity to review the revised manuscript. I appreciate the time and effort the authors put towards the revisions, and they have addressed many of the concerns raised in the first review. I have a few additional suggestions to strengthen the manuscript, mostly based on the writing in the introduction and discussion sections, to highlight the contributions of the study.

> Thank you for your positive feedback and additional suggestions on our work. We have further revised the introduction and discussion sections to better highlight the contributions of the study.

The revisions to the introduction provide a stronger rationale for the article, but the flow and organization could be improved. For example:

> Thanks for the detailed suggestions. Please see our point-by-point responses to address your specific comments below.

- The paragraph starting on line 35 (Given the broad interpretation of infrastructure) helps the define key terms, but it might be better placed later in the introduction, following the rationale and premise of the study. For instance, the previous paragraph ends with a statement about "significant knowledge gaps" on health and infrastructure, and some elaboration on these gaps would be helpful as the next paragraph before introducing the study (and how the study addresses these gaps).

> Following your suggestions, the paragraph on key term definitions related to infrastructure has been placed later in the introduction, and the entire section has been reorganized as follows:

- Paragraph 1 (Lines 42-54) introduces the significance of infrastructure in promoting sustainable development and human health, emphasizing the urgent need for research on their relationships to inform policymaking.
- Paragraphs 2–4 (Lines 55-100) elaborate existing knowledge gaps in detail, specifically concluded in three aspects: (1) quantification of human access to infrastructure, (2) evaluation of infrastructure access inequality, and (3) associations between infrastructure access and human health.

- Paragraph 5 (Lines 101-117) outlines the overarching goal of this research, defines key terms, and introduces the conceptual framework for economic-social-environmental infrastructure.
- Paragraph 6 (Lines 118-128) provides an overview of the methodologies used and key research questions addressed.

We believe this reorganization enhances the clarity and structure of the introduction, and presents the study's rationale and contributions in a clear and compelling logic.

- the article could more clearly lay out the knowledge gaps, directly link the literature reviews to the knowledge gaps, and then directly state the two key contributions of the study. Based on my reading, I see these contributes as 1) comparing infrastructural inequalities by country and 2) measuring health impacts of infrastructure inequalities.

> Thank you for your suggestion. In Paragraphs 2-4, based on our thorough literature review, we outlined existing research gaps regarding the limited understanding of global inequalities in infrastructure access and their health implications, noting that prior studies often focus on single infrastructure systems or regional-scale analyses. Building upon these discussions, Paragraph 6 begins by explicitly highlighting our study's motivation and contribution: *"To bridge these knowledge gaps, this paper aims to provide a comprehensive, multidimensional analysis of global inequality in infrastructure access and the associated disparity in health outcomes."* (Lines 101-103)

- the following sentence on line 90 seems too broad and sweeping, yet is critical to the research contribution of the study: "Furthermore, while access to infrastructure has been extensively studied, few studies have explored how inequality in infrastructure access correlates with health outcomes." There is substantial research on inequities in access to public health infrastructure and health outcomes, to water infrastructure inequities and health outcomes, etc. This sentence needs to be revised and the associated literature review needs to be refined.

> Thank you for pointing this out. We have revised the sentence to avoid confusion and incorporated relevant literature on the association between infrastructure access inequality (such as water and health care infrastructure) and health outcomes. The revised text now reads: *"Understanding how disparities in infrastructure access translate into health inequality is crucial, as infrastructure inequities often reflect and reinforce broader health disparities^{36, 46, 47}. Recent research has demonstrated that inequities in access to*

health care and water infrastructure can lead to increased health burdens^{48, 49}. Nonetheless, most studies examining the relationship between infrastructure access and human health focus on single infrastructure types, be they economic, social, or environmental, while research on health disparities across multiple infrastructure types often remains confined to regional scales^{46, 50}.” (Lines 93-100)

The discussion section is improved, and the addition of the limitations section is helpful. I recognize that this study is an exciting, large-scale, global study of infrastructure across multiple types of infrastructure, and I especially appreciate the focus on equitable access to infrastructure, however, based on the discussion section, the results are not particularly novel and they seem fairly intuitive (equitable access to infrastructure is associated with positive health outcomes, or see line 297 "Key infrastructure components, such as access to transportation, healthcare facilities, and housing, are essential in creating environments that are conducive to good living conditions"). Each of the points raised in the discussion are described as reflecting previous research. Can the authors better highlight the novel contribution of this study? What can this study tell us that previous work did not?

> Thank you for your recognition and valuable suggestions regarding research advancements. Following your suggestion on fleshing out our research novelty, we have revised the discussion section to better articulate how our results advance and complement current knowledge. The study's contributions are now summarized as follows, corresponding to Paragraphs 1–4 in the discussion: (1) identifying contrasting disparities in infrastructure access and inequality between the Global North and Global South (Lines 225-242); (2) informing global infrastructure inequality through a comprehensive assessment of multi-dimensional and cross-regional infrastructure access differences (Lines 243-260); (3) developing a novel approach to measuring infrastructure access by characterizing human-infrastructure interactions (Lines 261-278); and (4) building on points (1–3), advancing existing knowledge with empirical evidence on the associations between infrastructure access inequality and human health (Lines 279-292).

To explicitly highlight these contributions, we have incorporated statements such as:

- “Our findings **address this gap** by ...” (Line 227)
- “Another **contribution** of this study lies in ...” (Line 243)
- “The **advantages** of our framework lie in ...” (Line 248)
- “Our analysis also **sheds light** on ...” (Lines 279)
- “Our mixed-effects regression modelling further **advances current knowledge** by ...” (Lines 287-288)

Additionally, we have revised the text to avoid overly intuitive language, and the sentence you mentioned now reads as: *“While the provision of infrastructure services is fundamental to human health, addressing disparities in access, especially in regions with low overall infrastructure access, is even more vital. Our findings reveal that even small increases in its inequality, particularly in the economic dimension, can compromise health outcomes. This necessitates the need for targeted efforts to lessen inequalities in key infrastructure components such as transportation, telecommunications, and housing.”* (Lines 323-328)

We believe these revisions in terms of specific points to the discussion section further clarify and highlight the novelty and contributions of this study.

Reviewer #2 (Remarks to the Author):

The authors have done a thorough job of addressing feedback and improving the robustness of various aspects of the study. I am satisfied that all comments have now been addressed.

> Thank you for your kind feedback. We sincerely appreciate your time and thoughtful comments in improving the manuscript.

Reviewer #3 (Remarks to the Author):

The authors have done a thorough job of addressing my concerns. I have no further comments.

> Thank you for recognizing our work and taking the time to review the manuscript.

Remarks on code availability:

I am unsure what this is in reference to but I did see the authors provided sources for the data they used and descriptions of this data.

> Yes, all the data used in this study are sourced from open-access databases or cited literature, and they have been properly referenced in the manuscript. In addition, we have deposited all resulting data and the code used to produce the major findings of this study in an open repository (https://figshare.com/projects/Infrastructure_inequality/237854), which supports the reproducibility of our analysis.